# Efficient tandem electroreduction of nitrate into ammonia through coupling Cu single atoms with adjacent $Co_3O_4$

Yan Liu[1,7], Jie Wei[1,7], Zhengwu Yang[1,7], Lirong Zheng[2], Jiankang Zhao[1], Zhimin Song[1], Yuhan Zhou [1], Jiajie Cheng [3], Junyang Meng [1], Zhigang Geng [1] ✉ & Jie Zeng [1,4,5,6] ✉

The nitrate ($NO_3^-$) electroreduction into ammonia ($NH_3$) represents a promising approach for sustainable $NH_3$ synthesis. However, the variation of adsorption configurations renders great difficulties in the simultaneous optimization of binding energy for the intermediates. Though the extensively reported Cu-based electrocatalysts benefit $NO_3^-$ adsorption, one of the key issues lies in the accumulation of nitrite ($NO_2^-$) due to its weak adsorption, resulting in the rapid deactivation of catalysts and sluggish kinetics of subsequent hydrogenation steps. Here we report a tandem electrocatalyst by combining Cu single atoms catalysts with adjacent $Co_3O_4$ nanosheets to boost the electroreduction of $NO_3^-$ to $NH_3$. The obtained tandem catalyst exhibits a yield rate for $NH_3$ of 114.0 $mg_{NH_3}$ $h^{-1}$ $cm^{-2}$, which exceeds the previous values for the reported Cu-based catalysts. Mechanism investigations unveil that the combination of $Co_3O_4$ regulates the adsorption configuration of $NO_2^-$ and strengthens the binding with $NO_2^-$, thus accelerating the electroreduction of $NO_3^-$ to $NH_3$.

As one of the nitrogen-containing species, nitrate ($NO_3^-$) widely exists in industrial and agricultural wastewater with a high concentration, mainly caused by the emission of low-level nuclear waste and intensive usage of fertilizers[1-3]. Excessive $NO_3^-$ has significantly threatened ecological balance, inducing acid rain and photochemical smog[4]. Additionally, $NO_3^-$ in human body is easily converted into toxic nitrite ($NO_2^-$), leading to serious health issues[5]. Among the methods for removing $NO_3^-$, electroreduction process using renewable electricity is regarded as an appealing technology under mild conditions[6-9]. The controllable products including nontoxic nitrogen ($N_2$) and valuable ammonia ($NH_3$) could be obtained after $NO_3^-$ electroreduction[10-12].

Since $NH_3$ is a fundamental chemical compound and a promising green hydrogen carrier, $NO_3^-$ electroreduction into $NH_3$ instead of $N_2$ is more desirable. Taken together, it is highly imperative to achieve efficient electroreduction of $NO_3^-$ into $NH_3$ from the perspective of environmental protection and sustainable $NH_3$ synthesis.

In view of the multiple nitrogen-containing intermediates (e.g. $*NO_3$, $*NO_2$, and $*NO$) involved in the $NO_3^-$ electroreduction, an optimal catalyst should satisfy the simultaneously optimized adsorption of intermediates. The moderate binding energy of intermediates serves as one of the key factors for efficient $NO_3^-$ electroreduction into $NH_3$[13,14]. Classically, given that the coordination of N atom in $NO_3^-$ is

¹Hefei National Research Center for Physical Sciences at the Microscale, University of Science and Technology of China, Hefei 230026 Anhui, PR China. ²Institute of High Energy Physics, Chinese Academy of Sciences, 100049 Beijing, PR China. ³Department of Physics, University of Science and Technology of China, Hefei 230026 Anhui, PR China. ⁴CAS Key Laboratory of Strongly-Coupled Quantum Matter Physics, University of Science and Technology of China, Hefei 230026 Anhui, PR China. ⁵Key Laboratory of Surface and Interface Chemistry and Energy Catalysis of Anhui Higher Education Institutes, Department of Chemical Physics, University of Science and Technology of China, Hefei 230026 Anhui, PR China. ⁶School of Chemistry & Chemical Engineering, Anhui University of Technology, Ma'anshan 243002 Anhui, PR China. ⁷These authors contributed equally: Yan Liu, Jie Wei, Zhengwu Yang. ✉e-mail: gengzg@ustc.edu.cn; zengj@ustc.edu.cn

saturated by three O atoms, *NO$_3$ tends to bond with active sites through O atoms. Whereas, *NO$_2$ is preferentially adsorbed on active sites through N and O atoms. As for *NO, N atom in *NO is inclined to connect with active sites. The variation of adsorption configurations renders great difficulties in the simultaneous optimization of binding energy for the intermediates. A typical instance is Cu-based electrocatalysts which have been reported extensively for NO$_3^-$ electroreduction[14–20]. Though Cu-based electrocatalysts benefit NO$_3^-$ adsorption, one of the key issues lies in the accumulation of NO$_2^-$, resulting in the rapid deactivation of catalysts and sluggish kinetics of the subsequent hydrogenation steps for NH$_3$ production[15,17]. However, it remains a grand challenge to design an efficient catalyst to satisfy the simultaneously optimized adsorption of intermediates with different configurations.

Herein, we report a tandem electrocatalyst by combining Cu single atoms anchored on N-doped carbon with adjacent Co$_3$O$_4$ nanosheets (denoted as Co$_3$O$_4$/Cu$_1$-N-C) to boost the electroreduction of NO$_3^-$ to NH$_3$. The obtained Co$_3$O$_4$/Cu$_1$-N-C catalyst exhibits a remarkable yield rate for NH$_3$ of 114.0 mg$_{NH_3}$ h$^{-1}$ cm$^{-2}$, which exceeds the previous values for all of the reported Cu-based catalysts. Mechanism investigations unveil that the combination of Co$_3$O$_4$

regulates the adsorption configuration of NO$_2^-$ and strengthens the binding with NO$_2^-$, thus accelerating the electroreduction of NO$_3^-$ to NH$_3$.

## Results
### Catalyst synthesis and characterizations
Co$_3$O$_4$/Cu$_1$-N-C catalyst was synthesized by adding sodium borohydride to the mixture containing Cu single-atom catalysts and cobalt nitrate. Cu single atoms dispersed on N-doped carbon (denoted as Cu$_1$-N-C) were prepared via pyrolyzing Cu-doped ZIF-8 at 900 °C under Ar atmosphere (Supplementary Figs. 1 and 2). Figure 1a shows the high-angle annular dark field scanning transmission electron microscopy (HAADF-STEM) image of Co$_3$O$_4$/Cu$_1$-N-C. As displayed in the high resolution TEM (HRTEM) image and the corresponding selected area electron diffraction pattern (SAED), Co$_3$O$_4$ nanosheets were successfully deposited on the surface of Cu$_1$-N-C (Supplementary Fig. 3). Figure 1b shows the aberration-corrected HAADF-STEM image of Co$_3$O$_4$/ Cu$_1$-N-C. The lattice fringes with an interplanar spacing of 0.201 nm were ascribed to the (400) facet of Co$_3$O$_4$. Besides, abundant Cu single atoms were observed around Co$_3$O$_4$ nanosheets. Based on energy-dispersive X-ray spectroscopy (EDS) elemental mapping, Co, Cu, and N

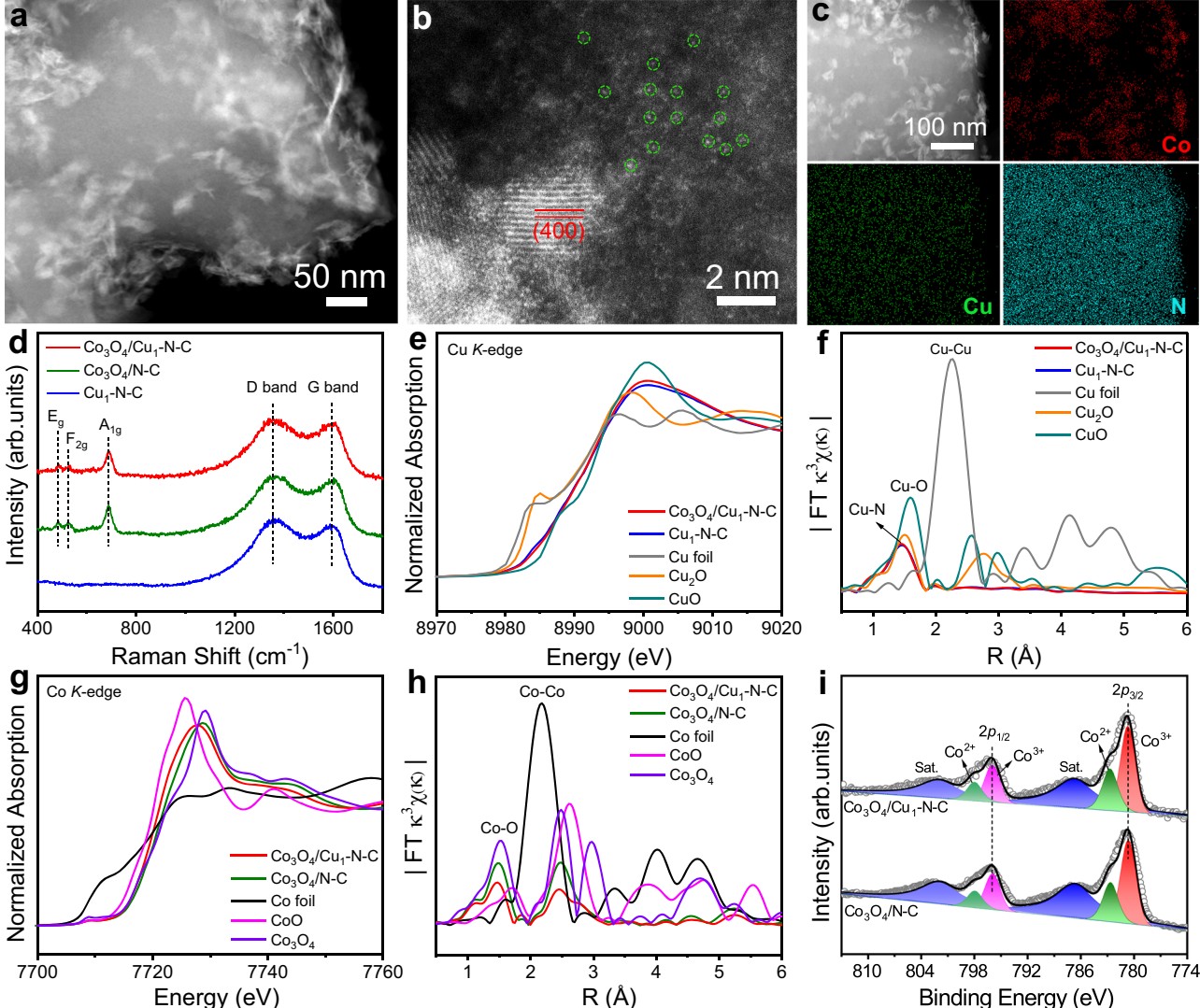

**Fig. 1 | Structural characterizations. a** HAADF-STEM image, **b** aberration-corrected HAADF-STEM image, and **c** EDS elemental mappings of Co$_3$O$_4$/Cu$_1$-N-C. **d** Raman spectra of Co$_3$O$_4$/Cu$_1$-N-C, Cu$_1$-N-C, and Co$_3$O$_4$/N-C. **e** Cu *K*-edge XANES spectra and **f** EXAFS spectra for Co$_3$O$_4$/Cu$_1$-N-C, Cu$_1$-N-C, Cu foil, Cu$_2$O, and CuO. **g** Co *K*-edge XANES spectra and **h** EXAFS spectra for Co$_3$O$_4$/Cu$_1$-N-C, Co$_3$O$_4$/N-C, Co foil, CoO, and Co$_3$O$_4$. **i** Co 2*p* XPS spectra for Co$_3$O$_4$/Cu$_1$-N-C and Co$_3$O$_4$/N-C.

elements were uniformly distributed throughout the whole structure (Fig. 1c). The uniform distribution of Cu sites and $Co_3O_4$ species constituted the adjacent catalytic centers. The metal content of Cu and Co in $Co_3O_4/Cu_1$-N-C were determined to be 0.60 wt% and 4.70 wt%, respectively, by inductively coupled plasma-optical emission spectroscopy analysis (ICP-OES). For comparison, $Co_3O_4$ nanosheets dispersed on N-doped carbon (denoted as $Co_3O_4$/N-C) were prepared with a similar synthetic procedure of $Co_3O_4/Cu_1$-N-C except for the addition of Cu precursor (Supplementary Fig. 4). Figure 1d shows the Raman spectra for $Co_3O_4/Cu_1$-N-C, $Cu_1$-N-C, and $Co_3O_4$/N-C. All of the Raman spectra displayed two peaks located at 1356 and 1591 $cm^{-1}$, assigned to the D band and G band of graphite carbon, respectively[21]. The similar intensity ratios of D band to G band ($I_D/I_G$) for the three samples indicated that the carbon support possessed similar degree of structural disorder (Supplementary Fig. 5). Compared with $Cu_1$-N-C, three distinguishable peaks located at 482, 527, and 689 $cm^{-1}$ were observed for both $Co_3O_4/Cu_1$-N-C and $Co_3O_4$/N-C, corresponding to $E_g$, $F_{2g}$, and $A_{1g}$ vibration modes of $Co_3O_4$ crystals, respectively[22]. The structure of graphite carbon supports was further confirmed by X-ray diffraction patterns (Supplementary Fig. 6). Figure 1e shows the Cu K-edge X-ray absorption near edge structure (XANES) spectra of $Co_3O_4/Cu_1$-N-C and $Cu_1$-N-C. Obviously, the energy absorption edge profiles for both $Co_3O_4/Cu_1$-N-C and $Cu_1$-N-C were located between those of CuO and $Cu_2O$, elucidating that the valence state of Cu species in the two catalysts were between +1 to +2. As shown in Fig. 1f, a dominant peak at 1.93 Å was observed in the extended X-ray absorption fine structure (EXAFS) spectra of Cu K-edge for $Co_3O_4/Cu_1$-N-C and $Cu_1$-N-C, which were attributed to the Cu-N bond. The absence of Cu-Cu bond in the two catalysts further confirmed the atomic dispersion of Cu species. Besides, the EXAFS fitting results indicate that the coordination numbers of Cu-N shell in both $Co_3O_4/Cu_1$-N-C and $Cu_1$-N-C were approximately 4.0 (Supplementary Fig. 7 and Table 1). After the deposition of $Co_3O_4$ nanosheets, the coordination structure of Cu single atoms ($CuN_4$) in $Co_3O_4/Cu_1$-N-C was unchanged. Besides, the wavelet transformed EXAFS (WT-EXAFS) spectra of $Co_3O_4/Cu_1$-N-C and $Cu_1$-N-C also confirmed the Cu-N bonding in the two catalysts (Supplementary Fig. 8). For the Co K-edge XANES spectra, the edge energy for both $Co_3O_4/Cu_1$-N-C and $Co_3O_4$/N-C were similar to that for $Co_3O_4$ reference (Fig. 1g). Figure 1h shows that the Co-O coordination in $Co_3O_4/Cu_1$-N-C and $Co_3O_4$/N-C were approximate to that in $Co_3O_4$, certifying the similar coordination structure of $Co_3O_4$ species in the two catalysts (Supplementary Fig. 9 and Table 2). Figure 1i shows the Co 2p X-ray photoelectron spectroscopy (XPS) spectra. Specifically, the peaks at 798.0, 782.7, 796.1, and 780.8 eV in $Co_3O_4/Cu_1$-N-C and $Co_3O_4$/N-C corresponded to $Co^{2+}$ $2p_{1/2}$, $Co^{2+}$ $2p_{3/2}$, $Co^{3+}$ $2p_{1/2}$, and $Co^{3+}$ $2p_{3/2}$, respectively[23]. The indiscernible shift of Co 2p peaks demonstrated that $Cu_1$-N-C as the support did not significantly affect the valence state of Co.

## Catalytic performance toward $NO_3^-$ electroreduction

The catalytic performance of the catalysts was investigated in a three-electrode H-type cell toward $NO_3^-$ electroreduction (Supplementary Fig. 10). The concentration of $NH_3$ product was quantified by the indophenol blue method (Supplementary Fig. 11). To preliminarily explore the process of tandem catalysis, we conducted the linear sweep voltammetry (LSV) curves of $Co_3O_4/Cu_1$-N-C, $Cu_1$-N-C, and $Co_3O_4$/N-C with 1 M $NO_3^-/NO_2^-$, respectively. As shown in Fig. 2a, the current density of $Cu_1$-N-C in the presence of $NO_3^-$ was higher than that of $Co_3O_4$/N-C, suggesting that $Cu_1$-N-C possessed higher activity toward $NO_3^-$ electroreduction. Whereas, $Co_3O_4$/N-C exhibited a larger current density relative to $Cu_1$-N-C in $NO_2^-$ (Fig. 2b). The superior activity of $Co_3O_4$ species toward $NO_2^-$ electroreduction was further demonstrated by the higher Faradaic efficiency (FE) and yield rate for $NH_3$ of $Co_3O_4$/N-C in $NO_2^-$ electroreduction relative to $Cu_1$-N-C (Supplementary Fig. 12). Considering that $NO_2^-$ is one of the vital

intermediates, the combination of $Cu_1$-N-C and $Co_3O_4$ would couple the separate functions of Cu sites and $Co_3O_4$ species for the sequential reduction of $NO_3^-$ and $NO_2^-$. As expected, $Co_3O_4/Cu_1$-N-C displayed the highest current density among the three catalysts in the electrolyte of $NO_3^-$. Besides, the tremendous discrepancy of the LSV curves of $Co_3O_4/Cu_1$-N-C in 1 M KOH with/without $NO_3^-$ also implied the superior activity of $Co_3O_4/Cu_1$-N-C toward $NO_3^-$ electroreduction (Supplementary Fig. 13).

Figure 2c provides the partial current density for $NH_3$ ($j_{NH3}$) of $Co_3O_4/Cu_1$-N-C, $Cu_1$-N-C, and $Co_3O_4$/N-C at various applied potentials toward $NO_3^-$ electroreduction. The $j_{NH3}$ of $Co_3O_4/Cu_1$-N-C exceeded those of $Cu_1$-N-C and $Co_3O_4$/N-C. Especially, at −1.0 V vs reversible hydrogen electrode (RHE), the $j_{NH3}$ of $Co_3O_4/Cu_1$-N-C reached −1437.5 mA $cm^{-2}$, which was 2.2 times and 3.6 times as high as that of $Cu_1$-N-C and $Co_3O_4$/N-C, respectively. Moreover, the normalized $j_{NH3}$ based on double-layer capacitance ($C_{dl}$) for $Co_3O_4/Cu_1$-N-C was the largest among the three catalysts, indicating the highest intrinsic activity for $Co_3O_4/Cu_1$-N-C toward $NO_3^-$ electroreduction (Fig. 2d and Supplementary Fig. 14). In addition, the FE for $NH_3$ of $Co_3O_4/Cu_1$-N-C was higher with respect to the other two counterparts at all applied potentials (Fig. 2e). Especially, $Co_3O_4/Cu_1$-N-C achieved the maximum FE for $NH_3$ of 97.7% at −0.8 V vs RHE. Furthermore, at −1.0 V vs RHE, the yield rate of $NH_3$ for $Co_3O_4/Cu_1$-N-C reached up to 114.0 $mg_{NH_3}$ $h^{-1}$ $cm^{-2}$, which exceeded all of the reported value for Cu-based catalysts[14,16,18,19,24–30] (Fig. 2f and Supplementary Table 3). The yield rate of $NH_4^+$ in the electrolyte after the electroreduction process was also determined by $^1H$ nuclear magnetic resonance (NMR) analysis, which was approximated to the results detected via the indophenol blue method (Supplementary Fig. 15 and Table 4). Other liquid and gaseous products including $NO_2^-$, $NH_2OH$, NO, $NO_2$, $N_2O$, $H_2$, and $N_2$ for $Co_3O_4/Cu_1$-N-C were also measured (Supplementary Figs. 16–18). $NH_3$ were the only main product after $NO_3^-/NO_2^-$ electroreduction (Supplementary Table 5). Besides, the FE for $NH_3$ of $Co_3O_4/Cu_1$-N-C with $NO_3^-$ concentrations ranging from 10 mM to 500 mM all exceeded 91.2%, indicating a wide tolerance range for the concentration of $NO_3^-$ (Supplementary Fig. 19). The durability of $Co_3O_4/Cu_1$-N-C was examined by 20 rounds of successive reactions. The negligible decay of the yield rate demonstrated the satisfactory durability of $Co_3O_4/Cu_1$-N-C (Fig. 2g). The Raman and XAFS measurements for $Co_3O_4/Cu_1$-N-C after the electrolysis indicated that the $Co_3O_4$ species and Cu-N bonding were preserved (Supplementary Figs. 20 and 21). The stability of Cu single atoms in $Co_3O_4/Cu_1$-N-C during the electrolysis was further explored by in situ EXAFS measurements, indicating that Cu atoms remained the atomically dispersed state in $Co_3O_4/Cu_1$-N-C during the $NO_3^-$ electroreduction (Supplementary Figs. 22 and 23).

To further clarify the synergy effect of $Co_3O_4$ on the conversion of $NO_2^-$, we conducted a series of control experiments. The catalytic performance of other metal oxides (such as $FeO_x$, $CuO_x$, and $NiO_x$) dispersed on N-doped carbon toward $NO_2^-$ electroreduction were all lower than that over $Co_3O_4$/N-C, suggesting the inferior ability of these metal oxide to facilitate $NO_2^-$ reduction (Supplementary Figs. 24 and 25). In addition, the loading amount of $Co_3O_4$ on $Cu_1$-N-C was vital to the efficient conversion of the accumulated $NO_2^-$ (Supplementary Fig. 26). Besides, the simply physical mixing of $Cu_1$-N-C and $Co_3O_4$/N-C could not sufficiently assure the spatial couple of the adjacent sites, thereby limiting the effective hydrogenation of $NO_2^-$ into $NH_3$ during $NO_3^-$ electroreduction (Supplementary Fig. 27). We also exclude the possible ammonia contamination from the self-electrolysis of $Co_3O_4/Cu_1$-N-C, electrolyte, and carbon paper, respectively (Supplementary Fig. 28). Besides, the catalytic activity of N-doped carbon was much lower compared with that of $Co_3O_4/Cu_1$-N-C (Supplementary Fig. 29). The possible interference of Co single atoms on $Cu_1$-N-C support could be considered insignificant to the catalytic performance of $Co_3O_4/Cu_1$-N-C (Supplementary Figs. 30–33). The electroreduction of $NO_3^-$ was also affected by the diffusion of

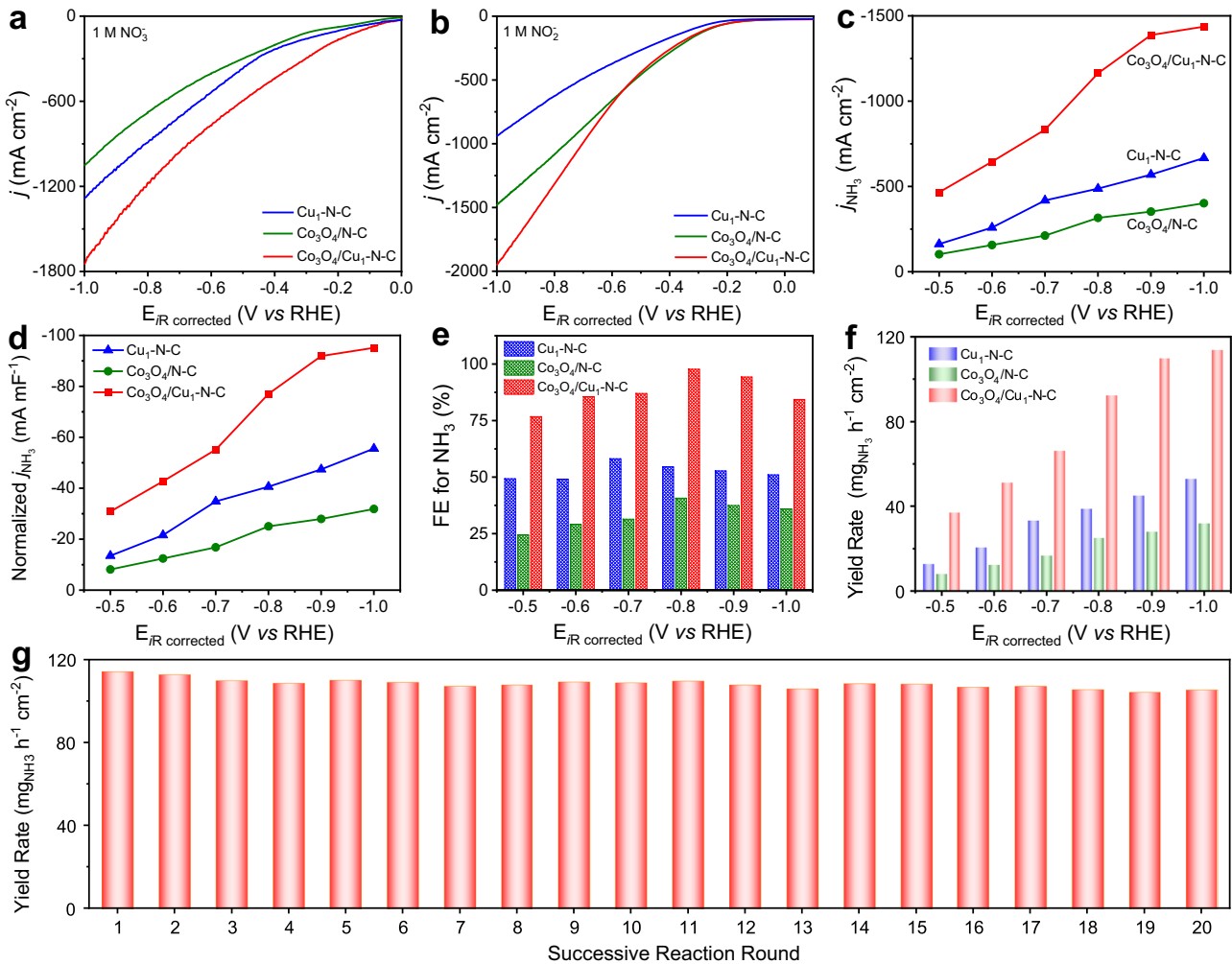

**Fig. 2 | Catalytic performance.** LSV curves of Cu$_1$-N-C, Co$_3$O$_4$/N-C, and Co$_3$O$_4$/Cu$_1$-N-C with (**a**) 1 M NO$_3^-$ and (**b**) 1 M NO$_2^-$. **c** $j_{NH_3}$, **d** normalized $j_{NH_3}$ based on $C_{dl}$, **e** FE for NH$_3$, and **f** yield rate for NH$_3$ of Cu$_1$-N-C, Co$_3$O$_4$/N-C, and Co$_3$O$_4$/Cu$_1$-N-C at different applied potentials with 1 M NO$_3^-$. **g** Yield rate for NH$_3$ of Co$_3$O$_4$/Cu$_1$-N-C at −1.0 V $vs$ RHE under 20 rounds of successive reactions. The solution resistance was determined to be $4.4 \pm 0.2$ ohm in the electrolytes by potentiostatic electrochemical impedance spectroscopy.

reactants (Supplementary Fig. 34). Moreover, $^{15}$NO$_3^-$ isotopic labeling measurements for Co$_3$O$_4$/Cu$_1$-N-C was conducted with $^1$H NMR analysis. Only typical doublet peaks attributed to $^{15}$NH$_4^+$ were collected with $^{15}$NO$_3^-$ as the N source whereas the triplet peaks of $^{14}$NH$_4^+$ were detected with $^{14}$NO$_3^-$ as the N source (Supplementary Fig. 35). These results indicated that the NH$_3$ detected in the electrolyte originated from the electroreduction of NO$_3^-$.

**Mechanistic study on NO$_3^-$ electroreduction**

To gain more insight into the catalytic process of NO$_3^-$ electroreduction over Co$_3$O$_4$/Cu$_1$-N-C, we conducted in situ electrochemical Fourier transform infrared spectroscopy (FTIR) and Raman spectroscopy to monitor the reaction process (Supplementary Figs. 36 and 37). Figure 3a displays the in situ FTIR spectra of Co$_3$O$_4$/Cu$_1$-N-C at applied potentials from OCP to −1.0 V $vs$ RHE. The negative peaks at 1382 cm$^{-1}$ were ascribed to the consumption of NO$_3^-$ species[3]. In addition, the emergence of positive peaks located at 1456 cm$^{-1}$ confirmed that NH$_4^+$ was generated during the NO$_3^-$ electroreduction[31]. Besides, two peaks at 1541 and 1508 cm$^{-1}$ were detected, which were assigned to the vibration band of *NO and *NOH, respectively (Supplementary Fig. 38). Figure 3b shows the in situ Raman spectra of Co$_3$O$_4$/Cu$_1$-N-C at all applied potentials. The peaks corresponding to E$_g$, F$_{2g}$, and A$_{1g}$ vibration modes of Co$_3$O$_4$ remained unchanged, suggesting that the Co$_3$O$_4$

species was stable during NO$_3^-$ electroreduction. During the NO$_3^-$ electroreduction, only the peak at 1049 cm$^{-1}$ was observed for Co$_3$O$_4$/Cu$_1$-N-C at all applied potentials, assigned to the symmetric stretching vibration of NO$_3^-$ (Fig. 3c). In the case of Cu$_1$-N-C, apart from the signal of NO$_3^-$, a new peak at 810 cm$^{-1}$ ascribed to the bending vibration of NO$_2^-$ gradually appeared as the applied potential increased, indicating the accumulation of NO$_2^-$ for Cu$_1$-N-C during NO$_3^-$ electroreduction (Fig. 3d). To further probe the variation of local concentration for NO$_2^-$ near the surface of the catalysts, we designed a Raman cell that allows Raman laser to detect from the surface of catalysts to the electrolyte bulk. The Raman laser was designed to be incident from the back of catalysts to diminish the interference from the strong absorbance of NO$_3^-$ in the electrolyte. As illustrated in Fig. 3e, the electrocatalysts were deposited on fluorine tin oxide-coated glass (FTO) as the working electrode (WE). The distance from the laser beam to electrode surface is controlled by the mechanical sample stage. Figure 3f, g display the in situ Raman spectra of Co$_3$O$_4$/Cu$_1$-N-C and Cu$_1$-N-C at −0.8 V $vs$ RHE when the laser beam was positioned 0 to 200 μm away from the surface of catalysts, respectively. With the increment of the distance between the focal plane of laser and the surface of catalysts, the signal intensity of graphite carbon for the catalysts gradually decreased. A noticeable peak at 810 cm$^{-1}$ assigned to NO$_2^-$ arose near the surface of Cu$_1$-N-C. Whereas, the signal of NO$_2^-$ for Co$_3$O$_4$/Cu$_1$-N-C was negligible,

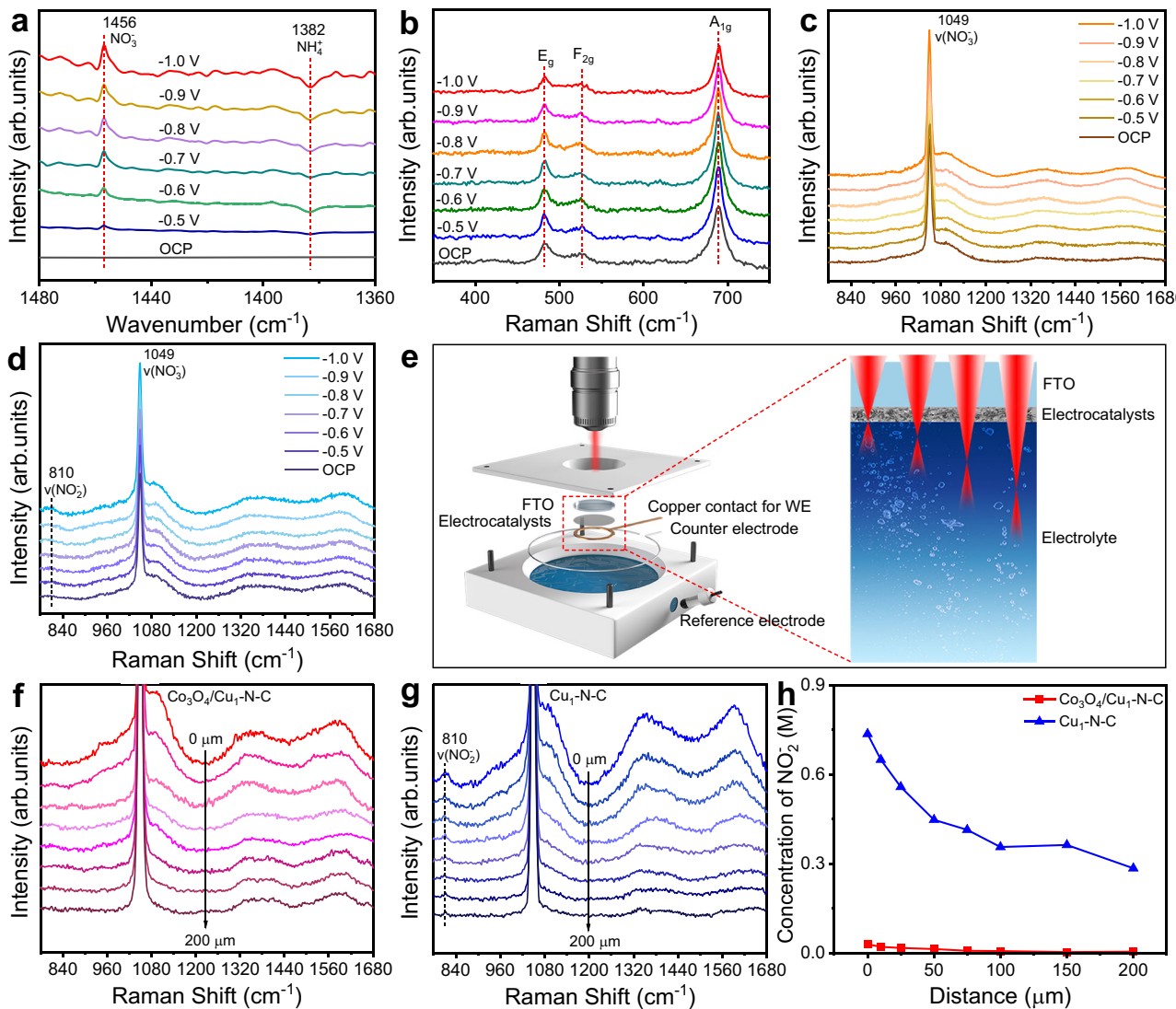

**Fig. 3 | In situ characterizations. a** In situ FTIR spectra for Co$_3$O$_4$/Cu$_1$-N-C from OCP to −1.0 V *vs* RHE in 1 M NO$_3^-$. **b** In situ Raman spectra for Co$_3$O$_4$/Cu$_1$-N-C from OCP to −1.0 V *vs* RHE in 1 M NO$_3^-$. In situ Raman spectra for (**c**) Co$_3$O$_4$/Cu$_1$-N-C and (**d**) Cu$_1$-N-C from OCP to −1.0 V *vs* RHE in 1 M NO$_3^-$. **e** Scheme of the designed Raman cell for detecting from the surface of catalysts to the electrolyte bulk. In situ Raman spectra for (**f**) Co$_3$O$_4$/Cu$_1$-N-C and (**g**) Cu$_1$-N-C at −0.8 V *vs* RHE in 1 M NO$_3^-$ with different distances ranging from 0 to 200 μm. **h** The calculated concentration of NO$_2^-$ for Co$_3$O$_4$/Cu$_1$-N-C and Cu$_1$-N-C with different distances ranging from 0 to 200 μm.

which was independent of the distance. Furthermore, we determined the local concentration of NO$_2^-$ near the surface of catalysts based on the integrated areas of NO$_2^-$ and NO$_3^-$, taking the ratio of integrated areas for 1 M NO$_2^-$ and 1 M NO$_3^-$ solutions as a correction factor (Supplementary Fig. 39). As the laser beam was set further far away from the surface of catalysts into the electrolyte, the concentration of NO$_2^-$ for Cu$_1$-N-C gradually decreased from 0.74 to 0.29 M (Fig. 3h). This trend indicates that the NO$_2^-$ generated at Cu$_1$-N-C/electrolyte interface diffused into the electrolyte due to the sluggish reduction of NO$_2^-$. Clearly, the concentration of NO$_2^-$ for Co$_3$O$_4$/Cu$_1$-N-C was much lower relative to Cu$_1$-N-C, manifesting the facilitated reduction of NO$_2^-$ with the favor of Co$_3$O$_4$.

To further understand the synergetic role of Cu$_1$-N-C and Co$_3$O$_4$ in the catalytic process, we calculated the rate constants for NO$_3^-$ electroreduction ($k_1$) and NO$_2^-$ electroreduction ($k_2$), respectively (Fig. 4a). The concentration of residual NO$_3^-$ after the electroreduction process was quantified by UV-Vis spectrophotometry (Supplementary Fig. 40). Compared with Co$_3$O$_4$/N-C, the larger $k_1$ value of Cu$_1$-N-C suggests that Cu$_1$-N-C was more favorable for the conversion of NO$_3^-$ to NO$_2^-$, but the

smaller $k_2$ value shows the slower kinetics for NO$_2^-$ reduction. Accordingly, excessive NO$_2^-$ would be desorbed into the electrolyte for Cu$_1$-N-C. Notably, the highest $k_1$ and $k_2$ of Co$_3$O$_4$/Cu$_1$-N-C manifested the simultaneous acceleration of the conversion of NO$_3^-$ to NO$_2^-$ and NO$_2^-$ to NH$_3$. Besides, the tafel slopes of Co$_3$O$_4$/Cu$_1$-N-C, Co$_3$O$_4$/N-C, and Cu$_1$-N-C in 1 M NO$_3^-$ and 1 M NO$_2^-$ imply that the combination of Co$_3$O$_4$ with Cu$_1$-N-C facilitate the kinetics of NO$_3^-$ and NO$_2^-$ reduction during the catalytic process (Supplementary Fig. 41). Figure 4b shows the adsorption capacities ($q_e$) of Co$_3$O$_4$/Cu$_1$-N-C, Cu$_1$-N-C, and Co$_3$O$_4$/ N-C for NO$_3^-$ and NO$_2^-$, respectively. It is obvious that Co$_3$O$_4$/Cu$_1$-N-C exhibited the largest $q_e$ for both NO$_3^-$ and NO$_2^-$ among the three catalysts. As a consequence, combining Cu$_1$-N-C with Co$_3$O$_4$ was conducive to the conversion of both NO$_3^-$ and NO$_2^-$.

The density functional theory (DFT) calculations were conducted to further interpret the reaction mechanism of NO$_3^-$ electroreduction over Co$_3$O$_4$/Cu$_1$-N-C catalysts. Based on the results of structural analysis, we adopted CuN$_4$ and Co$_3$O$_4$ (100) slabs as the models to calculate the Gibbs free energies ($G$) for each step involved in NO$_3^-$ electroreduction, respectively (Supplementary Fig. 42). After the

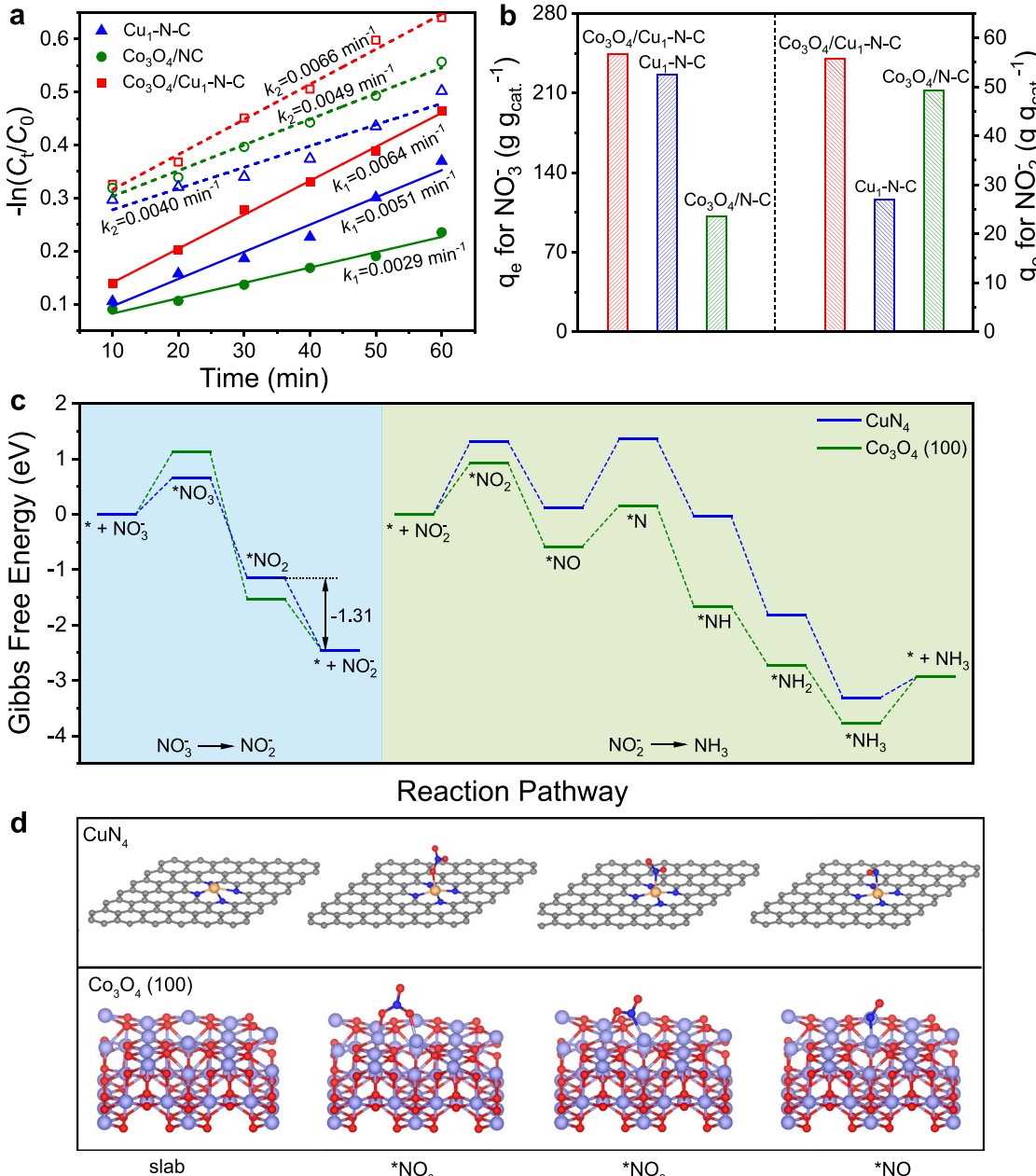

**Fig. 4 | Mechanistic study on NO₃⁻ electroreduction. a** Linearized pseudo first-order kinetic profiles of Co₃O₄/Cu₁-N-C, Cu₁-N-C, and Co₃O₄/N-C in 1 M NO₃⁻/NO₂⁻, respectively. **b** Adsorption capacities of Co₃O₄/Cu₁-N-C, Cu₁-N-C, and Co₃O₄/N-C for NO₃⁻/NO₂⁻. **c** Free energy diagram of NO₃⁻ electroreduction over CuN₄ and Co₃O₄ (100) slabs. * represents an adsorption site. **d** Structure models of key intermediates on CuN₄ and Co₃O₄ (100) slabs. The gray, blue, red, yellow, and purple spheres represent C, N, O, Cu, and Co atoms, respectively.

structure optimization, *NO₃ would be adsorbed on CuN₄ with O atom. As presented in Fig. 4c, the Gibbs free-energy changes (ΔG) for NO₃⁻ adsorption over CuN₄ is much lower than that over Co₃O₄ (100), indicating the stronger binding of NO₃⁻ over CuN₄. Nevertheless, the ΔG of *NO₂ desorption over CuN₄ is −1.31 eV, which could be more thermodynamically favorable than the reduction of *NO₂ to *NO (−1.20 eV). In this regard, the desorption of *NO₂ would give rise to the accumulation of NO₂⁻ over CuN₄, which was consistent with the high yield for NO₂⁻ over Cu₁-N-C in NO₃⁻ electroreduction (Supplementary Fig. 17). As for the conversion of NO₂⁻, *NO₂ would be adsorbed on CuN₄ through N atom whereas *NO₂ could be connected with Co₃O₄ (100) through N and O atoms after the structure optimization, leading to the lower ΔG for NO₂⁻ adsorption over Co₃O₄ (100) (Fig. 4d). In addition, the relative high ΔG for *H adsorption over CuN₄ and Co₃O₄

revealed the weak adsorption of *H, indicating that the occurrence of competitive H₂ evolution could be depressed (Supplementary Figs. 43 and 44). As a result, Co₃O₄ would regulate the adsorption configuration of NO₂⁻ and possess an easier binding with NO₂⁻, facilitating the reduction of NO₂⁻ to NH₃.

## Discussion
In summary, we developed a highly efficient catalyst by coupling the separate functions of Cu₁-N-C and Co₃O₄ for the sequential reduction of NO₃⁻ to NO₂⁻ and NO₂⁻ to NH₃. The obtained Co₃O₄/Cu₁-N-C catalyst exhibited a superior yield rate for NH₃ of 114.0 mg_NH₃ h⁻¹ cm⁻², which exceeded all of the reported values for Cu-based catalysts. The mechanism investigations unveiled that the combination of Co₃O₄ regulated the adsorption configuration of NO₂⁻ and strengthened the

binding with $NO_2^-$, thus accelerating the electroreduction of $NO_3^-$ to $NH_3$. This work offers a novel guideline for the construction of highly efficient tandem catalysts toward $NO_3^-$ electroreduction.

## Methods

### Chemicals and materials

Zinc nitrate hexahydrate ($Zn(NO_3)_2 \cdot 6H_2O$, 99.0%), 2-methyl imidazole (2-MeIM, 99.0%), copper(II) acetate monohydrate ($Cu(COOCH_3)_2 \cdot H_2O$, 99.0%), cobalt nitrate hexahydrate ($Co(NO_3)_2 \cdot 6H_2O$, 99.0%), iron nitrate nonahydrate ($Fe(NO_3)_3 \cdot 9H_2O$, 98.5%), nickel nitrate hexahydrate ($Ni(NO_3)_2 \cdot 6H_2O$, 98.0%), copper nitrate trihydrate ($Cu(NO_3)_2 \cdot 3H_2O$, 99.0%), methanol (99.5%), ethanol (99.5%), potassium nitrate ($KNO_3$, 99.0%), potassium nitrite ($KNO_2$, 97.0%), potassium hydroxide (KOH, 85%), ammonium sulfate ($(NH_4)_2SO_4$, 99.0%), sodium hydroxide (NaOH, ≥96.0%), salicylic acid ($C_7H_6O_3$), sodium hypochlorite solution (NaClO, available chlorine 5.2% of aqueous solution), trisodium citrate dihydrate ($C_6H_5Na_3O_7 \cdot 2H_2O$), sodium nitroferricyanide dihydrate ($C_5FeN_6Na_2O \cdot 2H_2O$), hydrochloric acid (HCl, 12 mol $L^{-1}$), sulfamic acid (99.5%), p-aminobenzenesulfonamide (98.0%), N-(1-Naphthyl) ethylenediamine dihydrochloride (98.0%), phosphoric acid ($H_3PO_4$, ≥85.0%), and 1-propanesulfonic acid 3-(trimethylsilyl) sodium salt (DSS) were purchased from Sinopharm Chemical Reagent Co. Ltd. Glyoxylic acid solution ($C_2H_2O_3$, 50 wt%), dimethyl sulfoxide-$d_6$ (DMSO-$d_6$, 99.9 atom% D), ($^{15}NH_4)_2SO_4$ (99.0 atom% $^{15}N$), and $K^{15}NO_3$ (99.0 atom% $^{15}N$) were purchased from Aladdin Chemistry Co., Ltd (Shanghai, China). Bipolar membrane (TRJBM) were purchased from Beijing Tingrun Membrane Technology Development Co., Ltd (Beijing, China). The deionized (DI) water was produced using a Millipore Milli-Q grade, with a resistivity of 18.2 MΩ cm. All of the chemicals were used without any further purification.

### Instrumentations

TEM images were taken using a Hitachi HT7700 transmission electron microscope at an acceleration voltage of 100 kV. HAADF-STEM and the corresponding EDS elemental mapping were carried out on a Talos F200X field-emission transmission electron microscope operated at an accelerating voltage of 200 kV using Mo-based TEM grids. Aberration-corrected HAADF-STEM images were carried out on Themis Z field-emission transmission electron microscope operating at an accelerating voltage of 300 kV using Mo-based TEM grids. XRD patterns were collected using a Rikagu MiniFlex X-ray diffractometer with Cu-Kα radiation (λ = 1.54059 Å). ICP-OES (Avio 220 MAX, PerkinElmer) analysis was employed to measure the concentration of metal species. XPS measurements were performed using a Kratos Axis supra+ diffractometer with Al-Kα radiation. The Raman spectra were conducted via LabRAM HR Evolution (Horiba) Raman system with a 532 nm excitation laser. The absorbance data was measured on a UV-vis spectrophotometer (Agilent Technologies, Cary 60). The in situ FTIR spectra were acquired by a Nicolet iS50 FTIR spectrometer with a built-in MCT detector.

### Synthesis of Cu₁-N-C

A mixture of $Zn(NO_3)_2 \cdot 6H_2O$ (5.6 mmol) and $Cu(COOCH_3)_2 \cdot H_2O$ (0.28 mmol) was dissolved in 80 mL of methanol, which was subsequently added into 80 mL of methanol containing 3.70 g of 2-MeIM. Then the mixed solution was kept at 25 °C for 12 h. The as-obtained precipitate (denoted as Cu-doped ZIF-8) was separated by centrifugation and washed subsequently with methanol for five times, and finally dried at 65 °C under vacuum overnight. Next, the obtained Cu-containing derivative of ZIF-8 was heated to 900 °C with a heating rate of 5 °C $min^{-1}$ in a tube furnace and kept at 900 °C under flowing Ar gas for 3 h. After the tube furnace was naturally cooled to room temperature, Cu₁-N-C was obtained and directly used as the catalyst without further treatment. For comparison, Co single atoms anchored on N-doped carbon (denoted as Co₁-N-C) and Co single atoms

anchored on Cu₁-N-C (denoted as Co₁Cu₁-N-C) were obtained via pyrolyzing the Co-doped ZIF-8 and Cu/Co-doped ZIF-8, respectively. Co-doped ZIF-8 and Cu/Co-doped ZIF-8 prepared with the similar procedure with that of Cu₁-N-C except the $Co(NO_3)_2 \cdot 6H_2O$ and the mixture of $Co(NO_3)_2 \cdot 6H_2O$ and $Cu(COOCH_3)_2 \cdot H_2O$ as the metal precursors, respectively.

### Synthesis of Co₃O₄/Cu₁-N-C and Co₃O₄/N-C

160 mg of Cu₁-N-C was dispersed in 20 mL of ethanol by sonication for 30 min. Afterwards, 5 mL of $H_2O$ containing 45 mg of $Co(NO_3)_2 \cdot 6H_2O$ was added into the above solution, which was maintained in an ice-water bath for 1 h with vigorous stirring. Then, 40 mL of freshly prepared $NaBH_4$ (100 mg) with ice-cold $H_2O$ was added dropwise into the above suspension, followed by further stirring for 1 h. The as-obtained precipitate was separated by filtration and washed subsequently with water for five times. Finally, Co₃O₄/Cu₁-N-C was obtained by being dried at 65 °C under vacuum overnight. Co₃O₄/N-C was prepared as a comparison with the similar procedure with that of Co₃O₄/Cu₁-N-C except for the addition of N-doped carbon instead of Cu₁-N-C. N-doped carbon was prepared with a similar synthetic procedure with that of Cu₁-N-C without the addition of $Cu(COOCH_3)_2 \cdot H_2O$. For comparison, other metal oxides including $FeO_x$, $CuO_x$, and $NiO_x$ dispersed on N-doped carbon (denoted as $FeO_x$/N-C, $CuO_x$/N-C, and $NiO_x$/N-C, respectively) were prepared with the similar procedure with that of Co₃O₄/N-C except for the addition of $Fe(NO_3)_3 \cdot 9H_2O$, $Cu(NO_3)_2 \cdot 3H_2O$, and $Ni(NO_3)_2 \cdot 6H_2O$ as the metal precursors, respectively (denoted as $FeO_x$/N-C, $CuO_x$/N-C, and $NiO_x$/N-C, respectively).

### X-ray absorption fine structure (XAFS) measurements

The XAFS spectra at Cu K-edge and Co K-edge were performed at 1W1B beamline of Beijing Synchrotron Radiation Facility and BL11B beamline of Shanghai Synchrotron Radiation Facility. The data were obtained in ambient conditions under fluorescence mode for Cu K-edge and transmission mode for Co K-edge, respectively.

The ATHENA module and ARTEMIS codes in the IFEFFIT software packages were employed to extract the data and fitted the profiles[32–34]. The $k^3$-weighted EXAFS spectra were acquired by energy calibration and spectral normalization. For the EXAFS part, the Fourier transformed data in R space of Cu K-edge and Co K-edge were analyzed by applying a hanning windows (d$k$ = 1.0 $Å^{-1}$) to differentiate the EXAFS oscillation from different coordination shells. Subsequently, we performed the least-squares curve parameter fitting to attain the structural parameters around central atoms. The fitted ranges of $k$ space were set at 3.4–13.2 $Å^{-1}$ with R range of 1.2–3.0 Å. The four parameters including coordination number (CN), bond length (R), Debye-Waller factor ($\sigma^2$), and $E_0$ shift ($\Delta E_0$) were fitted without anyone being fixed, constrained, or correlated.

The in situ Cu K-edge XAFS measurements were conducted were collected with a home-made XAFS cell. Typically, 8 mg of the catalysts and 40 μL of Nafion were dispersed in 2 mL of ethanol by sonication for 1 h. Then the uniform ink was loaded onto carbon paper with an area of 2 × 2 $cm^2$. The mass loading was calculated to be 2 mg $cm^{-2}$. The prepared catalysts, a Ag/AgCl electrode, and a Pt wire were used as the working electrode, reference electrode, and counter electrode, respectively. All electrochemical tests were measured in 1 M KOH electrolyte with 1 M $KNO_3$ (45 mL) and controlled by a CHI1140C electrochemical workstation.

### Preparation of the working electrodes

8 mg of the catalysts were dispersed in 2 mL of ethanol by sonication for 1 h. Then 40 μL of Nafion solution was added to the mixture and sonicated for 30 min to obtain a uniform ink. Finally, the uniform ink was loaded onto carbon paper with an area of 2 × 4 $cm^2$. The mass loading was calculated to be 1 mg $cm^{-2}$. The area of working electrodes used in the electrochemical measurements was 0.25 $cm^2$.

## Electrochemical measurements

The electrochemical measurements were carried out in an H-cell system which was separated by a bipolar membrane with a CHI1140C electrochemical workstation (Chenhua, Shanghai). Ag/AgCl electrode and graphite rod were used as the reference electrode and counter electrode, respectively. For $NO_3^-$ electroreduction, 1 M KOH containing 1 M $KNO_3$ solution (60 mL) was evenly distributed to the cathode and anode compartments. The pH value of the electrolyte was determined to be 14 by a FiveEasy Plus pH Meter (METTLER TOLEDO). The applied potentials were measured against the Ag/AgCl reference electrode with 50% $i$R compensation and converted to the RHE reference scale by E ($vs$ RHE) = E ($vs$ Ag/AgCl) + 0.21 V + 0.0591 × pH − $i$R. The solution resistance was determined to be 4.4 ± 0.2 ohm in the electrolytes by potentiostatic electrochemical impedance spectroscopy at frequencies ranging from 10 Hz to 100 kHz, which was conducted in a standard three-electrode system at ambient conditions. Before the electroreduction test, CV curves were performed until the polarization curves achieved steady-state ones with a scan rate of 10 mV s$^{-1}$. Before the electrolysis, Ar gas was delivered into the cathodic compartment at a rate of 10 mL min$^{-1}$ to remove dissolved $O_2$. The LSVs of the catalysts were recorded at a scan rate of 5 mV s$^{-1}$ in 1 M KOH containing 1 M $KNO_3$/$KNO_2$. The controlled potential electrolysis was performed at applied potentials for 10 min. $NO_2^-$ electroreduction was conducted with the same conditions except that the solution of 1 M KOH containing 1 M $KNO_2$ was used as the electrolyte. An absorption cell containing 30 mL of 1 M HCl was set to effectively absorb the possible escaped $NH_3$ from the cathode cell. After the electrolysis at each applied potential, the concentration of $NH_3$ in the absorption cell was lower than 1 µg mL$^{-1}$. In this case, the volatilization of $NH_3$ from the electrolytes could be negligible. Cyclic voltammetric measurements were conducted in a non-faradaic potential window with various scan rates from 50 to 100 mV s$^{-1}$. $C_{dl}$ was calculated by plotting the $\Delta j$ ($\Delta j = j_a - j_c$) at the middle of the corresponding potential window against scan rates. The $j_a$ and $j_c$ were the anodic and cathodic current densities, respectively. The slope was twice of $C_{dl}$.

## The calculation method for FE

The FE for the product ($NH_3$ and $NO_2^-$) was calculated at a given potential as follows:

$$FE = C \times V \times N \times F/(Q \times M) \tag{1}$$

$C$: the measured concentration of product (mg mL$^{-1}$),
$V$: the volume of the electrolyte (mL),
$N$: the number of electrons transferred for the product, which is 8 for $NH_3$ and 2 for $NO_2^-$,
F: Faraday constant, 96,485 C mol$^{-1}$,
$Q$: total electric charge (C),
M: the relative molecular mass, which is 17 g mol$^{-1}$ for $NH_3$ and 46 g mol$^{-1}$ for $NO_2^-$.

## The calculation method for the yield rate of $NH_3$ product

The yield rate of $NH_3$ product was calculated at a given potential as follows:

$$\nu_{NH_3} = (C_{NH_3} \times V)/(S \times t) \times 60 \tag{2}$$

$v_{NH_3}$: the yield rate (mg$_{NH_3}$ h$^{-1}$ cm$^{-2}$),
$C_{NH_3}$: the measured concentration of $NH_3$ (mg mL$^{-1}$),
$V$: the volume of the electrolyte (mL),
$S$: the area of the catalyst (cm$^2$),
$t$: the reduction reaction time (min).

## Determination of ion concentration

Determination of $NH_3$ concentration with indophenol blue method[35]. After the electroreduction process, a certain amount of electrolyte was taken out from the electrolytic cell and diluted to the detection range. Then, 2 mL of 1 M NaOH solution containing salicylic acid (5 wt%) and sodium citrate (5 wt%) were added into the aforementioned solution, followed by the addition of 1 mL of 0.05 M NaClO and 0.2 mL of $C_5FeN_6Na_2O$ (1 wt%). After standing in darkness for 2 h, the absorption spectra were measured using a UV-vis spectrophotometer. The concentration of indophenol blue was determined using absorbance at the wavelength of 650 nm. The concentration-absorbance curve was calibrated using standard $(NH_4)_2SO_4$ solution with a series of concentrations.

Determination of $NH_3$ concentration with $^1$H NMR method. After $NO_3^-$ electroreduction, a certain amount of electrolyte was taken out for further quantification by $^1$H NMR (Bruker AVANCE AV III 400). All analyses were performed with 128-time scans. The concentration-integral area curve was calibrated using a standard $(NH_4)_2SO_4$ solution. Typically, $(NH_4)_2SO_4$ was dissolved in 20 mL of 1 M KOH electrolyte as a series of standard $(NH_4)_2SO_4$ solutions with different concentrations. Subsequently, 0.1 mL of dimethyl sulfoxide-$d_6$ (DMSO-$d_6$), 0.1 mL of 6 mM 1-propanesulfonic acid 3-(trimethylsilyl) sodium salt (DSS) solution, and 0.08 mL of 6 M HCl to adjust the pH value were added into 0.32 mL of $(NH_4)_2SO_4$ standard solutions with different concentrations. The signal appeared at 7.23, 7.10, and 6.97 ppm were attributed to $NH_4^+$. The integral areas of the signal of $NH_4^+$ were used to determine the concentration of $(NH_4)_2SO_4$ compared with the as-known DSS reference.

Determination of $NO_3^-$ concentration[36]. A certain amount of electrolyte was diluted to the detection range of $NO_3^-$. Then, 0.1 mL of 1 M HCl and 0.01 mL of 0.8 wt% sulfamic acid solution were mixed with the diluted electrolyte, followed by shaking for 10 min. Using a UV-vis spectrophotometer, the absorption spectra were collected, obtaining the absorption intensities at a wavelength of 220 and 275 nm. Finally, the calculated absorbance (A = A$_{220nm}$−2A$_{275nm}$) was acquired. The concentration-absorbance curve was calibrated using standard $KNO_3$ solutions with a series of concentrations.

Determination of $NO_2^-$ concentration. 4 g of p-aminobenzene-sulfonamide, 0.2 g of N-(1-Naphthyl) ethylenediamine dihydrochloride, and 10 mL of phosphoric acid were mixed with 50 mL of water as the color reagent. A certain amount of electrolyte was taken out from the electrolytic cell and diluted to the detection range. 1 mL of $H_3PO_4$ (5 M) was added to the 4 mL of diluted post-electrolysis electrolytes to adjust the pH, followed by the addition of 0.1 mL of color reagent. After standing for 20 min, the absorption spectra were measured using a UV-vis spectrophotometer. The absorption intensity at a wavelength of 540 nm was recorded. The concentration-absorbance curve was calibrated using standard $KNO_2$ solution with a series of concentrations.

## Determination of other liquid and gaseous product

The amount of $NH_2OH$ was determined by $^1$H NMR after $NH_2OH$ was captured by excess amount of $C_2H_2O_3$ through oximation process. Specifically, 0.4 mL of the electrolyte after the $NO_3^-$ electroreduction was mixed with 10 µL of 50% $C_2H_2O_3$ solution, followed by the addition of 0.1 mL of DMSO-$d_6$ (99%) and 0.1 mL of 6 mM DSS solution. The integral area of the signal appeared at 7.46 ppm were used to determine the concentration of $NH_2OH$ compared with the as-known DSS reference. In this work, the amount of $NH_2OH$ was below the detection limit for both $NO_3^-$ and $NO_2^-$ reduction over $Co_3O_4/Cu_1$-N-C.

Nitrogen oxides including NO, $NO_2$, and $N_2O$ have been detected by an infrared gas analyzer (THA100S). $H_2$ and $N_2$ have been detected by an on-line gas chromatograph (GC-2014) equipped with a flame ionization detector and a thermal conductivity detector.

The FE for gaseous products were calculated by the following equation:

$$FE = x \times V_{gas} \times N \times F/(Q \times V_m) \qquad (3)$$

$x$: the measured mole fraction of product,

$V_{gas}$: the total volume of the gas (L),

$N$: the number of electrons transferred for the product, which is 3 for NO, 8 for $N_2O$, 2 for $H_2$ and 10 for $N_2$,

F: Faraday constant, 96485 C mol$^{-1}$,

$Q$: total electric charge (C),

$V_m$: the molar volume of the gas, 24.5 L mol$^{-1}$.

## Isotope labeling experiments

The isotopic labeling experiment used K$^{15}$NO$_3$ with $^{15}$N enrichment of 99% as the feeding N-source to clarify the source of ammonia. 1 M KOH was used as the electrolyte and K$^{15}$NO$_3$ with a concentration of 1 M was added into the cathode compartment as the reactant. After the electrolysis, 0.1 mL of DMSO-$d_6$ and 0.1 mL of 6 mM DSS solution were added into 0.4 mL of the electrolyte, followed by adding 0.05 mL of HCl (0.1 M) to adjust the pH of the solutions. Then the obtained $^{15}$NH$_4^+$ was identified on a Varian 400 MHz NMR spectrometer (Bruker AVANCE AV III 400).

## Kinetic evaluation

The electrolysis at −1.0 V *vs* RHE were conducted for different time to acquire the rate constant in 1 M KOH containing 1 M NO$_3^-$ or 1 M NO$_2^-$. The reaction constant ($k_1$ for NO$_3^-$ reduction and $k_2$ for NO$_2^-$ reduction) was calculated by plotting the concentration of NO$_3^-$ or NO$_2^-$ against the time of reaction, supposing that the concentrations of NO$_3^-$ or NO$_2^-$ declined exponentially as per first-order rate.

$$C_t = C_0 \exp(-k \times t) \qquad (4)$$

$C_0$: initial concentration of NO$_3^-$ or NO$_2^-$ (g mL$^{-1}$),

$C_t$: the concentration of NO$_3^-$ or NO$_2^-$ at time $t$ (g mL$^{-1}$),

$t$: the time of reaction (min).

## Adsorption experiments

To determine the adsorption capacities of Co$_3$O$_4$/Cu$_1$-N-C, Cu$_1$-N-C, and Co$_3$O$_4$/N-C, 5 mg of catalysts were added to each 25 mL of NO$_3^-$ or NO$_2^-$ solutions with the initial concentration of 1 M under stirring for 2 h, respectively. The solutions were separated by filtration using the 0.22 μm microporous membrane filter. For high concentration of NO$_3^-$ or NO$_2^-$, the solution was diluted before absorbance measurements. The adsorption capacity was calculated using the following equation:

$$q_e = (C_0 - C_e) \times V/m \qquad (5)$$

q$_e$: the adsorption capacity (g g$_{cat.}^{-1}$),

$C_0$: the initial concentration of NO$_3^-$ or NO$_2^-$ (g mL$^{-1}$),

$C_e$: the measured concentration of NO$_3^-$ or NO$_2^-$ after the adsorption (g mL$^{-1}$),

$V$: the volume of the electrolyte (mL),

$m$: the mass of the catalyst (g).

## In situ FTIR measurements

Using a Nicolet iS50 FTIR spectrometer (Thermo Scientific) with a built-in MCT detector, we obtained the in situ electrochemical FTIR spectra. Typically, 2 mg of catalysts and 20 μL of Nafion dispersed in 2 mL of ethanol were sonicated for 1 h. Then the mixture was loaded onto the Au-coated Si prism to completely cover the Au film. The prepared prism was used as the working electrode after being dried naturally. The reference electrode and counter electrode was a Ag/ AgCl electrode and a Pt wire, respectively. The photograph of the in situ FTIR electrochemical cell was shown in Supplementary Fig. 36. All electrochemical tests were measured in 1 M KOH electrolyte with 0.1 M KNO$_3$ (30 mL) and controlled by a CHI1140C electrochemical workstation. All experiments were conducted at room temperature. The background spectra of the working electrode were obtained at an open-circuit potential before the electrochemical tests. All of the spectra were collected in absorbance by averaging 32 scans at a resolution of 4 cm$^{-1}$.

## In situ Raman measurements

In situ Raman was carried out using Lab RAM HR Evolution (Horiba) equipped with a 50× microscope objective. The excitation wavelength was 532 nm with 10% intensity. The photograph of electrochemical cell for in situ Raman measurement was shown in Supplementary Fig. 37. Typically, 2 mg of catalysts and 20 μL of Nafion were dispersed in 2 mL of ethanol by sonication for 1 h. Then the electrocatalysts were deposited on fluorine tin oxide-coated glass as the working electrode. A Ag/AgCl electrode, and a Pt wire were used as the reference electrode, and counter electrode, respectively. All electrochemical tests were measured in 1 M KOH electrolyte with 1 M KNO$_3$/KNO$_2$ (5 mL) and controlled by a CHI1140C electrochemical workstation. All experiments were conducted at room temperature. Each spectrum was collected by integration twice, 60 s per integration. To determine the local concentration of NO$_2^-$ near the surface of catalysts, we employed an internal standard method during the in situ Raman measurements.

## DFT calculations

DFT calculations were performed using the Vienna Ab-Initio Simulation Package (VASP) code at the GGA level within the PAW-PBE formalism[37]. DFT-D3 method with Becke-Jonson damping is performed for the van der Waals correction. The three-layer Co$_3$O$_4$ (100) slab model was adopted with a vacuum of 15 Å. The total energy calculations were performed using a 2 × 2 × 1 grid and a plane wave cut-off energy of 400 eV. Atoms in the bottom two layers were fixed. A U value of 3.5 eV was applied to the 3$d$ states of Co to describe the strong on-site Coulomb interactions due to the localization of the Co 3$d$ states[38]. For the model of CuN$_4$ (no atoms were fixed), the total energy calculations were performed using a 3 × 3 × 1 grid and a plane wave cut-off energy of 400 eV. All atoms, which were not fixed, including adsorbates were allowed to relax until the force on each ion was smaller than 0.02 eV/Å.

We calculated the Gibbs free energy ($G$) for each species as follows:

$$G = E_{DFT} + E_{ZPE} - TS \qquad (6)$$

where $E_{DFT}$, $E_{ZPE}$, and $TS$ represent the DFT-optimized total energy, zero point energy (ZPE), and entropy contribution, respectively (T is the temperature, 298.15 K). It is assumed that S = 0 for all the adsorbed species.

## Data availability

The data that support the findings of this study are available from the corresponding author upon request. The source data underlying Figs. 1–4 and Supplementary Figs. 1–44 are provided as a Source Data file. Source data are provided with this paper.

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

## Acknowledgements

This work was supported by the Strategic Priority Research Program of the Chinese Academy of Sciences (XDB0450401), National Key Research and Development Program of China (2021YFA1500500, 2019YFA0405600), NSFC (22209161, 22302184, 22322901, 22221003, 22250007, and 22361162655), National Science Fund for Distinguished Young Scholars (21925204), CAS project for young scientists in basic research (YSBR-022, YSBR-051), China Postdoctoral Program for Innovative Talents (BX20200324), Collaborative Innovation Program of Hefei Science Center, CAS (2022HSC-CIP004), International Partnership Program of Chinese Academy of Sciences (123GJHZ2022101GC), the Joint Fund of the Yulin University and the Dalian National Laboratory for Clean Energy (YLU-DNL Fund 2022012), Fundamental Research Funds for the Central Universities, the Anhui Natural Science Foundation for Young Scholars (2208085QB41), and the Fellowship of China Post-doctoral Science Foundation (2021M693058). J.Z. acknowledges support from the Tencent Foundation through the XPLORER PRIZE. This work was partially carried out at the Instruments Center for Physical

Science, University of Science and Technology of China. This work was also partially carried out at the USTC Center for Micro and Nanoscale Research and Fabrication.

## Author contributions

Z.G. and J.Zeng. supervised this project. Y.L. performed most of the experiments and analyzed the experimental data. J.W. conducted the in situ Raman measurements. Z.Y. and J.Zhao carried out DFT calculations and analyzed the computational data. L.Z. conducted the XAFS measurements and analyzed the results. Z.S., Y.Z., J.C., and J.M. provided help in materials synthesis and characterizations. Y.L., J.W., Z.G., and J.Zeng. wrote the manuscript. All authors discussed the results and assisted during manuscript preparation.

## Competing interests

The authors declare no competing interests.
