## [Peer Review File · Nature Communications]

REVIEWER COMMENTS

Reviewer #1 (Remarks to the Author):

This manuscript proposes a tandem catalyst design to address the issue of catalyst deactivation and slow kinetics of subsequent steps caused by the accumulation of NO₂⁻ in Cu-based catalysts. The authors characterized the catalytic mechanism using techniques such as in-situ infrared spectroscopy, in-situ Raman spectroscopy, and density functional theory (DFT) calculations. Ultimately, the prepared catalyst achieved a yield of 114.0 mgNH₃ h⁻¹ cm⁻². However, the concept explored in this study lacks novelty as previous investigations on the topic (e.g., <https://www.nature.com/articles/s41467-023-39366-9>; <https://www.nature.com/articles/s41467-022-35533-6>) have already been published. Therefore, it is not recommended for publication in Nature Communications.

The following suggestions are provided to enhance its quality:

1. It is preferable to compare the X-ray diffraction (XRD) data with the standard reference cards for enhanced analysis.
2. Has any alteration been observed in the I_d/I_g ratio of the Raman spectra depicted in Figure 1(d)?
3. It would be advantageous to provide wide-angle X-ray absorption fine structure (WT-EXAFS) data for further comprehensive examination.
4. The authors mentioned successful deposition of Co₃O₄ nanosheets on Cu-N-C surface, however, sheet-like structures are not distinctly discernible in the high-angle annular dark-field scanning transmission electron microscopy (HAADF-STEM) images.
5. It is recommended to employ ICP or other characterization techniques to determine the atomic content of Cu and Co₃O₄ loaded in the catalyst.
6. Considering the XAS and XPS results, it can be inferred from Figure 1(f) that there is complete overlap between Co₃O₄/Cu-N-C and Cu-N-C. This suggests that the introduction of Co₃O₄ does not affect the electronic distribution of Cu. The loading method of Co₃O₄ as well as how the presence of coupled adjacent catalytic centers are need clarification in order to understand the relationship between Co₃O₄ and Cu.
7. In Figure 1(i), it would be valuable to investigate if any band shifts can be observed in both Co₃O₄/Cu-N-C and Co₃O₄/N-C.
8. The author mentioned that "NO₂⁻ is one of the vital intermediates," but did not provide the yield of NO₂⁻ for each catalyst when examining their performance..
9. Is there a difference in catalytic performance between Co₃O₄/Cu-N-C and a physical mixture of Cu/N-C and Co₃O₄/N-C?
10. Further analysis is required for Figure 3(a). Are there any other intermediate products observed, such as nitrogen oxides?

11. In Figure 3(g)(h), was an internal standard incorporated during the in-situ Raman measurements to ensure the accuracy of the determined local NO₂⁻ concentration near the catalyst surface? Moreover, does the elevated concentration of NO₃⁻ during the Raman measurements impact the detection of NO₂⁻, NH₃, and other trace intermediates?
12. Do bimetallic sites facilitate kinetic transfer during catalytic processes? It is recommended to include Tafel slopes for each catalyst.
13. Employing two or more detection methods for product quantification to validate data accuracy is highly recommended.
14. The purpose of N doping in the catalyst remains unclear. Please provide an explanation.
15. Considering ammonia's solubility in alkali, was exhaust gas treatment conducted during the experimental process?
16. The article discusses the impact of NO₂⁻ accumulation on reaction rates. Was mechanical stirring employed during electrochemical testing? It is recommended to investigate this effect through control experiments.

Reviewer #2 (Remarks to the Author):

In this manuscript, Liu et al. report a tandem catalyst for ammonia synthesis from nitrate reduction by combining Cu single atoms and Co₃O₄ nanosheets both dispersed on nitrogen-doped carbon. The authors also employed in-situ tests and DFT calculations to clarify the synergetic role of Cu and Co₃O₄ sites, respectively, for the simultaneously optimized binding of nitrate and nitrite ions. At present, this work seems interesting for significant alleviation of nitrite production by the designed tandem process. I will recommend this manuscript for publication after some major revisions.

1. After reading the overall content and research in this manuscript, I feel the presented title is not appropriate for this whole work. The authors highlighted the “simultaneously optimized adsorption” of different reaction intermediates, but I have a doubt about how to optimize the adsorption simultaneously by the catalyst Co₃O₄/Cu-N-C. Firstly, only two reaction-related species of *NO₃ and *NO₂ were monitored by the combining in-situ spectra, as we know, the above both species are also presented in bulk electrolytes, it is not a real short-life intermediate during the whole reaction process. Besides, I cannot find the full explanation about the simultaneous optimization on the experimental spectra and theoretical calculations. In fact, a tandem mechanism for Cu-N-C and Co₃O₄ toward nitrate reduction is easier to understand for this work.
2. For this synthetic process of “by adding NaBH₄ to the mixture containing Cu-N-C and Co(NO₃)₂”, I think some Co atoms have also existed on Cu-N-C support as single atoms. As shown in Figure 1c, some Co mapping signals present in Cu-N-C support rather than Co₃O₄ aggregates.

3. In Figure 1b, could the authors provide an electron diffraction pattern to better support the structural formation of Co₃O₄ nanosheets?
4. The metal contents of Copper and Cobalt in Co₃O₄/Cu-N-C are suggested to be added.
5. To better clarify the structure of the catalysts, XAFS fitting at the Co K-edge for various samples should be provided.
6. In the section of activity measurements, the authors only tested one ammonia product, all FE values for both nitrate and nitrite reduction are less than one hundred percent. Are there additional gaseous and liquid products existing? These should be tested and reported.
7. Following the second comment, the possible effect of Co single atoms toward nitrate/nitrite reduction should be considered by the authors. Therefore, the ammonia activity of Co-N-C and Co single atoms on Cu-N-C catalysts also need test as a comparison.
8. In fact, after introducing Co₃O₄ to Cu-N-C catalyst, the total number of active sites should be increased. Does it mean the high activity partly originates from the high loading of Co₃O₄ in Co₃O₄/Cu-N-C? The activity comparison of physical-mixed Co₃O₄ and Cu-N-C catalysts should be provided.
9. After successive 20 electrolysis, the ammonia production seems decline in Figure 2h. Some examinations for spent catalyst should be measured.
10. The ammonia activity of N-C toward nitrate/nitrite reduction and LSV curves of Co₃O₄/Cu-N-C in KOH, respectively, should also be measured as a comparison.
11. The detected intermediate species on Raman and FTIR spectra should be labeled in Figures.
12. The more experimental details for in-situ measurements are suggested to show in the supplementary information.
13. In Figure 3, besides the only two detected species of *NO₃ and *NO₂ by combining Raman and FITR spectra, why are the other intermediates absent in this work? I cannot understand these spectral results, maybe some full explanations should be needed by the authors.
14. Considering the competitive HER during the electrolysis, the adsorption strength of hydrogen atom on the surface of catalysts is also important. The authors should calculate the free energy of hydrogen adsorption on Co₃O₄ and Cu-N-C, respectively.

Reviewer #3 (Remarks to the Author):

See attached

Manuscript Number: NCOMMS-23-47388

Article Type: Article

Title: Simultaneously optimized adsorption of intermediates in electrosynthesis of ammonia from nitrate through coupling Cu single atoms with adjacent Co₃O₄

Summary:

In this manuscript, authors promote the nitrate reduction reaction (NO₃RR) performance of Cu-based catalysts by combining Cu single atoms with Co₃O₄. NO₂⁻ is found to be the intermediate of NO₃RR. Cu-based electrocatalysts have poor binding affinity for NO₂⁻, resulting in the sluggish absorption and reduction of NO₂⁻. By dispersing Cu single atoms on Co₃O₄, the catalyst of Co₃O₄/Cu₁-N-C can be prepared and it is observed that the activity of NO₂⁻ reduction is highly improved. The FE and YR in electrosynthesis of ammonia from nitrate can reach 97.7% at -0.8 V vs RHE and 114.0 mgNH₃·h⁻¹·cm⁻² at -1.0 V vs RHE. The mechanism study further provides evidence that Co₃O₄ can facilitate NO₂⁻ binding which increases the catalytic NO₃RR activity. This manuscript is recommended for publication after addressing the following questions.

Here are some suggestions to improve the manuscript:

Major issues:

1. Could the authors perform the control experiments on N-doped carbon without the catalyst?
2. Why do authors choose Co₃O₄? Could the authors examine other metal oxides to compare their ability to bind intermediates?
3. Could the authors investigate how a change in the ratio between Cu single atom and Co₃O₄ would affect the overall NO₃RR performance?
4. Figure 2, why is the activity of Co₃O₄/Cu₁-N-C slightly lower than that of Co₃O₄/N-C when the potential is more negative than -0.5 V vs RHE?
5. Page 4, Line 93. The lattice fringes with an interplanar spacing of 0.201 nm were ascribed to the (400) facet of Co₃O₄. This number is different from reported Co₃O₄. Could the authors explain this difference? Does this difference have an effect on NO₂⁻ absorption?
6. Page 6, Line 143. Co₃O₄/Cu₁-N-C achieved the maximum FE for NH₃ of 97.7% at -0.8 V vs RHE. Could the authors quantify the other products under this condition, which could make up for the remaining FE? (e.g. NO₂⁻, NH₂OH, NO, NO₂, N₂O, N₂, and/or H₂)
7. The best yield rate is obtained at -1.0 V vs RHE. Could the authors comment on the competitive reaction (e.g. HER) under this (rather negative) applied potential?
8. Page 7, Line 190. The authors calculated the rate constants for NO₃⁻ and NO₂⁻ electroreduction (k_1 & k_2). In Figure S8, the author excluded the possible ammonia contamination from the self-electrolysis of the catalyst under NO₃⁻ conditions. Likewise, could the authors perform an experiment to exclude self-catalysis under NO₂⁻ conditions?
9. Page 16, Figure 3f and 3d, what are the two peaks around 1380 cm⁻¹ and 1620 cm⁻¹? Why do they look different for the two catalysts? Why do the signals change when the focal plane of the Raman laser is moved further away from the surface of the catalysts? Also,

why did the authors put emphasis on this laser-distance experiment? Is there any specific reason?

10. Page 17, Figure 4c. In order to compare the Gibbs Free Energy of CuN_4 and Co_3O_4 with the intermediate, the free energy diagram of NO_3^- and NO_2^- intermediate is separated into two parts with a blue and green background, and both begin at 0 eV. While the NO_3RR is a continuous process, is it reasonable to merge these two graphs into one? Could the authors explain more in-depth as to why the figure is drawn this way? Is it to show that CuN_4 (blue line) is the better catalyst for the NO_3^- -to- NO_2^- process (blue region) while Co_3O_4 (green line) is the better catalyst for the NO_2^- -to- NH_3 process (green region)?

Point-by-point response to reviewer comments

Manuscript ID: NCOMMS-23-47388

MS Type: Research Article

Title: “Efficient tandem electroreduction of nitrate into ammonia through coupling Cu single atoms with adjacent Co₃O₄”

Reviewer #1

This manuscript proposes a tandem catalyst design to address the issue of catalyst deactivation and slow kinetics of subsequent steps caused by the accumulation of NO₂⁻ in Cu-based catalysts. The authors characterized the catalytic mechanism using techniques such as in-situ infrared spectroscopy, in-situ Raman spectroscopy, and density functional theory (DFT) calculations. Ultimately, the prepared catalyst achieved a yield of 114.0 mg_{NH₃} h⁻¹ cm⁻². However, the concept explored in this study lacks novelty as previous investigations on the topic (e.g., <https://www.nature.com/articles/s41467-023-39366-9>; <https://www.nature.com/articles/s41467-022-35533-6>) have already been published. Therefore, it is not recommended for publication in Nature Communications.

We sincerely thank this reviewer for his/her careful reading of our manuscript and valuable comments to help us improve the quality of our manuscript. Despite that the dual sites in Cu-based catalysts have been reported to improve the performance for NO₃⁻ electroreduction, the intrinsic reason for the synergy effect in mentioned previous literatures are entirely different from that in our work. For example, the redistribution of electrons in Cu₅₀Co₅₀ alloy facilitated the electrons transfer and balanced the adsorption energy between *H and *NO₃ species (*Nat. Commun.*, **2022**, *13*, 7899). The addition of Co to Cu enhanced *H adsorption to improve the proton availability, thus promoting the hydrogenation during NO₃⁻ reduction. Besides, Fe/Cu diatomic catalyst presented the synergistic effects of dual atoms by lowering the energy barrier of NO₃⁻ adsorption and activating N-O bonds, leading to a high NH₃ yield and selectivity (*Nat. Commun.*, **2023**, *14*, 3634). The previously reported works mainly focused on the adsorption of NO₃⁻ and the supply of H for the hydrogenation step. However, for the extensively reported Cu-based catalysts, the undesired excessive accumulation of NO₂⁻ usually caused the deactivation of catalysts and limited the hydrogenation steps for NH₃ production. In this case, the tandem Cu-based catalysts require the combination of specific functional modules which could optimize the adsorption of NO₂⁻ intermediates. In our work, the key point lies in the investigation into the separate functions of Cu single atoms and Co₃O₄ toward the conversion of NO₃⁻ to NO₂⁻ and NO₂⁻ to NH₃, respectively. The NO₂⁻ generated on Cu sites was subsequently hydrogenated with the favor of Co₃O₄, which was evidenced by *in situ* Raman experiments, adsorption experiment, and theoretical calculation. The facilitated sequential reduction of NO₃⁻ to NO₂⁻ and NO₂⁻ to NH₃ through the tandem catalysis accelerated the electroreduction of NO₃⁻ to NH₃. This work not only develops an attractive Cu-based tandem catalyst for electroreduction of NO₃⁻ but also offers a legible profile for deepening the

understanding of tandem catalysis.

The following suggestions are provided to enhance its quality:

1. It is preferable to compare the X-ray diffraction (XRD) data with the standard reference cards for enhanced analysis.

We sincerely thank the reviewer for his/her valuable comments. In the revised manuscript, we have added the standard reference cards of Co_3O_4 and graphite carbon to compare with the X-ray diffraction (XRD) patterns of $\text{Co}_3\text{O}_4/\text{Cu}_1\text{-N-C}$, $\text{Co}_3\text{O}_4/\text{N-C}$, and $\text{Cu}_1\text{-N-C}$. As shown in Supplementary Figure 6, the three samples all exhibited two broad peaks at 24.5° and 44.0° , which were attributed to the (002) and (001) facets of graphite carbon (JCPDS No. 89-8487). However, the diffraction patterns of Co_3O_4 in $\text{Co}_3\text{O}_4/\text{Cu}_1\text{-N-C}$ and $\text{Co}_3\text{O}_4/\text{N-C}$ were not observable, which might be covered by the strong intensity of graphite carbon. We have added the corresponding discussions in the revised manuscript (page 4, lines 102-103, highlighted in yellow color).

2. Has any alteration been observed in the I_d/I_g ratio of the Raman spectra depicted in Figure 1(d)?

We sincerely thank the reviewer for his/her constructive comments. As shown in Supplementary Figure 5, the intensity ratio of D band to G band (I_D/I_G) of $\text{Co}_3\text{O}_4/\text{Cu}_1\text{-N-C}$ was calculated to be 1.11, which was approximate with those of $\text{Co}_3\text{O}_4/\text{N-C}$ (1.10) and $\text{Cu}_1\text{-N-C}$ (1.10). As such, the carbon support in the three samples possessed similar degree of structural disorder, suggesting that the loading method of Co_3O_4 had no obvious effect on the carbon structure. We have added the corresponding discussions in the revised manuscript (page 4, lines 98-100, highlighted in yellow color).

3. It would be advantageous to provide wide-angle X-ray absorption fine structure (WT-EXAFS) data for further comprehensive examination.

We appreciate the reviewer's valuable suggestion. We have provided the wavelet transformed EXAFS (WT-EXAFS) spectra of the $\text{Co}_3\text{O}_4/\text{Cu}_1\text{-N-C}$ and $\text{Cu}_1\text{-N-C}$ to further clarify the structure of Cu single atoms in the revised manuscript. As displayed in Supplementary Figure 8, the WT-EXAFS spectra of $\text{Co}_3\text{O}_4/\text{Cu}_1\text{-N-C}$ and $\text{Cu}_1\text{-N-C}$ both showed the maximum intensity at around 5.6 \AA^{-1} , demonstrating the Cu-N coordination structure in the two catalysts. For comparison, the maximum intensity of WT-EXAFS spectra for Cu foil at around 7.9 \AA^{-1} was ascribed to Cu-Cu bond. These results further confirmed the atomically dispersed Cu atoms on $\text{Co}_3\text{O}_4/\text{Cu}_1\text{-N-C}$ and $\text{Cu}_1\text{-N-C}$. We have added the corresponding discussions in the revised manuscript (page 5, lines 113-115, highlighted in yellow color).

4. The authors mentioned successful deposition of Co_3O_4 nanosheets on Cu-N-C surface, however, sheet-like structures are not distinctly discernible in the high-angle

annular dark-field scanning transmission electron microscopy (HAADF-STEM) images.

We genuinely thank the reviewer for raising this issue. The observation of Co_3O_4 sheets was indistinct in the original manuscript, which might be due to the low magnification of the HAADF-STEM image of $\text{Co}_3\text{O}_4/\text{Cu}_1\text{-N-C}$. To clearly show the sheet-structure of Co_3O_4 , we have added another HAADF-STEM image of $\text{Co}_3\text{O}_4/\text{Cu}_1\text{-N-C}$ with higher magnification in the revised manuscript. As shown in Supplementary Figure 3a, the sheet-like structures of Co_3O_4 could be observed through the whole catalyst. Besides, we have provided the typical high resolution transmission electron microscopy (HRTEM) image of $\text{Co}_3\text{O}_4/\text{Cu}_1\text{-N-C}$, presenting that the wrinkled nanosheets of Co_3O_4 were deposited on $\text{Cu}_1\text{-N-C}$ (Supplementary Fig. 3b). We have added the corresponding discussions in the revised manuscript (page 4, lines 83-84, highlighted in yellow color).

5. It is recommended to employ ICP or other characterization techniques to determine the atomic content of Cu and Co_3O_4 loaded in the catalyst.

As suggested, we have employed inductively coupled plasma-optical emission spectroscopy analysis (ICP-OES) to determine the content of Cu and Co. The metal content of Cu and Co in $\text{Co}_3\text{O}_4/\text{Cu}_1\text{-N-C}$ were determined to be 0.60 wt% and 4.70 wt%, respectively. We have added the corresponding discussions in the revised manuscript (Page 4, lines 91-93, highlighted in yellow color).

6. Considering the XAS and XPS results, it can be inferred from Figure 1(f) that there is complete overlap between $\text{Co}_3\text{O}_4/\text{Cu-N-C}$ and Cu-N-C . This suggests that the introduction of Co_3O_4 does not affect the electronic distribution of Cu. The loading method of Co_3O_4 as well as how the presence of coupled adjacent catalytic centers are need clarification in order to understand the relationship between Co_3O_4 and Cu.

We sincerely thank this reviewer for pointing out this issue. The loading of Co_3O_4 on $\text{Cu}_1\text{-N-C}$ was achieved through a wet-chemical method. Typically, $\text{Cu}_1\text{-N-C}$ catalyst were dispersed in ethanol by sonication, followed by the addition of $\text{Co}(\text{NO}_3)_2 \cdot 6\text{H}_2\text{O}$ with vigorous stirring for 8 h to completely adsorbed Co^{2+} ions. Then, NaBH_4 was added to make sure the fully reduction of Co^{2+} . The as-obtained $\text{Co}_3\text{O}_4/\text{Cu}_1\text{-N-C}$ was separated by filtration and washed with water, followed by being dried at 60 °C. The mild synthesis method of $\text{Co}_3\text{O}_4/\text{Cu}_1\text{-N-C}$ did not obviously affect the electronic distribution of Cu.

As we mentioned above, Co_3O_4 species were deposited on the surface of $\text{Cu}_1\text{-N-C}$, obtaining $\text{Co}_3\text{O}_4/\text{Cu}_1\text{-N-C}$ as a supported nano-composite catalyst. As displayed in Figure 1b, abundant Cu single atoms were observed around Co_3O_4 nanosheets. The uniform distribution of Cu sites and Co_3O_4 species throughout the whole structure constituted the adjacent catalytic centers for electroreduction of NO_3^- . The couple of adjacent Cu sites and Co_3O_4 species benefited the transfer of NO_2^- intermediates, promoting the hydrogenation step to NH_3 production. We have added the corresponding

discussions in the revised manuscript (Page 4, lines 90-91, highlighted in yellow color).

7. In Figure 1(i), it would be valuable to investigate if any band shifts can be observed in both $\text{Co}_3\text{O}_4/\text{Cu-N-C}$ and $\text{Co}_3\text{O}_4/\text{N-C}$.

We appreciate the reviewer's constructive suggestion. We have carefully checked the Co 2p XPS peaks in Figure 1i. The Co 2p XPS peaks of $\text{Co}_3\text{O}_4/\text{Cu}_1\text{-N-C}$ and $\text{Co}_3\text{O}_4/\text{N-C}$ showed no observable shift, demonstrating that $\text{Cu}_1\text{-N-C}$ as the support did not significantly affect the valence state of Co. As such, the electron transfer between Co_3O_4 and $\text{Cu}_1\text{-N-C}$ was rather weak to be ignored in $\text{Co}_3\text{O}_4/\text{Cu}_1\text{-N-C}$. We have added the corresponding discussions in the revised manuscript (Page 5, lines 122-123, highlighted in yellow color).

8. The author mentioned that " NO_2^- is one of the vital intermediates," but did not provide the yield of NO_2^- for each catalyst when examining their performance.

We genuinely thank this reviewer for the valuable comments. We have provided the yield rate of NO_2^- for $\text{Co}_3\text{O}_4/\text{Cu}_1\text{-N-C}$, $\text{Cu}_1\text{-N-C}$, and $\text{Co}_3\text{O}_4/\text{N-C}$ towards NO_3^- electroreduction in the revised Supplementary Information. As shown in Supplementary Figure 16, $\text{Cu}_1\text{-N-C}$ delivered the highest yield rate for NO_2^- among the three catalysts. Whereas, the yield rate for NO_2^- of $\text{Co}_3\text{O}_4/\text{Cu}_1\text{-N-C}$ was much lower than that of $\text{Cu}_1\text{-N-C}$. The excessive accumulation of NO_2^- for $\text{Cu}_1\text{-N-C}$ during the electroreduction of NO_3^- hampered the formation of NH_3 . Moreover, the composition of Co_3O_4 with $\text{Cu}_1\text{-N-C}$ facilitated the subsequent reduction of NO_2^- generated from $\text{Cu}_1\text{-N-C}$, thus accelerating the electroreduction of NO_3^- into NH_3 . We have added the corresponding discussions in the revised manuscript (Page 6, lines 156-158, highlighted in yellow color).

9. Is there a difference in catalytic performance between $\text{Co}_3\text{O}_4/\text{Cu-N-C}$ and a physical mixture of Cu/N-C and $\text{Co}_3\text{O}_4/\text{N-C}$?

We genuinely thank the reviewer for raising this issue. In the revised manuscript, we have evaluated the catalytic performance of a physical mixture of $\text{Cu}_1\text{-N-C}$ and $\text{Co}_3\text{O}_4/\text{N-C}$ (denoted as mixed $\text{Cu}_1+\text{Co}_3\text{O}_4$) in 1 M KOH with 1 M NO_3^- . As shown in Supplementary Figure 23, the FE for NH_3 of mixed $\text{Cu}_1+\text{Co}_3\text{O}_4$ was 71.4% at -0.8 V vs RHE, higher than those of $\text{Cu}_1\text{-N-C}$ (54.9%) and $\text{Co}_3\text{O}_4/\text{N-C}$ (40.9%) alone, indicating the combined contributions of $\text{Cu}_1\text{-N-C}$ and $\text{Co}_3\text{O}_4/\text{N-C}$ toward NO_3^- electroreduction. However, the catalytic performance of mixed $\text{Cu}_1+\text{Co}_3\text{O}_4$ was inferior compared with $\text{Co}_3\text{O}_4/\text{Cu}_1\text{-N-C}$ (97.7%). In this case, the spatial couple of the adjacent sites could not be sufficiently assured by simply physical mixing, thereby limiting the effective hydrogenation of NO_2^- into NH_3 during NO_3^- electroreduction. We have added the corresponding discussions in the revised manuscript (Page 7, lines 169-172, highlighted in yellow color).

10. Further analysis is required for Figure 3(a). Are there any other intermediate products observed, such as nitrogen oxides?

We genuinely thank the reviewer for raising this concern. In the original manuscript, we only focused on the consumption of NO_3^- and generated NH_4^+ in the *in situ* FTIR spectra of $\text{Co}_3\text{O}_4/\text{Cu}_1\text{-N-C}$, which were detected at the wavenumbers ranging from 1360 to 1480 cm^{-1} . To further identify the intermediates during the NO_3^- electroreduction, we have carefully analyzed the original data of the *in situ* FTIR spectra. As displayed in Supplementary Figure 34, two peaks at 1541 and 1508 cm^{-1} were observed at the wavenumbers ranging from 1490 to 1550 cm^{-1} , which were assigned to the vibration band of $^*\text{NO}$ and $^*\text{NOH}$, respectively. The intensities of the two peaks increased with the increment of applied potentials, implying the fast proceeding of NO_3^- electroreduction. We have added the corresponding discussions in the revised manuscript (Page 7, lines 190-192, highlighted in yellow color).

11. In Figure 3(g)(h), was an internal standard incorporated during the in-situ Raman measurements to ensure the accuracy of the determined local NO_2^- concentration near the catalyst surface? Moreover, does the elevated concentration of NO_3^- during the Raman measurements impact the detection of NO_2^- , NH_3 , and other trace intermediates?

We sincerely thank this reviewer for pointing out these issues. Indeed, we employed an internal standard method during the *in situ* Raman measurements to determine the local concentration of NO_2^- near the surface of catalysts in this work. Considering the distinct sensitivity of Raman measurements for NO_3^- and NO_2^- , the ratio of integrated areas for peaks of 1 M NO_3^- and 1 M NO_2^- was calculated as a correction factor. Then, the local concentration of NO_2^- near the catalyst surface were determined based on the integrated areas for the peaks of NO_3^- and NO_2^- during the electroreduction process. We have added the corresponding discussions in the revised Methods section (Page 18, lines 515-516, highlighted in yellow color).

As shown in Figures 3g and h, the peaks of NO_3^- and NO_2^- appeared at 1049 and 810 cm^{-1} , respectively. With such great difference in the peaks position, the high concentration of NO_3^- during the Raman measurements would not impact the detection of NO_2^- . However, the broad bands of graphite carbon from 1320 to 1680 cm^{-1} could overlap with the signals of other intermediates. Besides, the sensitivity of normal Raman is relatively low, limiting the detection of trace intermediates with low concentration. In this case, we focused on the detection of NO_2^- to probe the variation of local concentration for NO_2^- near the surface of the catalysts.

12. Do bimetallic sites facilitate kinetic transfer during catalytic processes? It is recommended to include Tafel slopes for each catalyst.

We genuinely thank the reviewer for his/her valuable comment. As suggested, we have analyzed the Tafel slopes based on the LSV curves of $\text{Co}_3\text{O}_4/\text{Cu}_1\text{-N-C}$, $\text{Co}_3\text{O}_4/\text{N-C}$, and

Cu₁-N-C in 1 M NO₃⁻ and 1 M NO₂⁻, respectively. As depicted in Supplementary Figure 37, in 1 M NO₃⁻, the Tafel slopes of Co₃O₄/Cu₁-N-C was 359 mV dec⁻¹, which was smaller than those of Co₃O₄/N-C (437 mV dec⁻¹) and Cu₁-N-C (550 mV dec⁻¹). This result implies that Co₃O₄/N-C facilitated the kinetics of NO₃⁻ reduction. In the case of 1 M NO₂⁻, Co₃O₄/Cu₁-N-C showed a Tafel slope of 439 mV dec⁻¹, close to that of Co₃O₄/N-C (427 mV dec⁻¹) but lower than that of Cu₁-N-C (844 mV dec⁻¹), indicating a faster NO₂⁻ reduction kinetics on Co₃O₄ species. As such, the combination of Co₃O₄ with Cu₁-N-C also promoted the kinetics of the subsequent NO₂⁻ reduction. We have added the corresponding discussions in the revised manuscript (Page 9, lines 228-230, highlighted in yellow color).

13. Employing two or more detection methods for product quantification to validate data accuracy is highly recommended.

We appreciate the reviewer's valuable suggestion. We have employed ¹H nuclear magnetic resonance (¹H NMR) to further determine the concentration of NH₄⁺ in the revised manuscript. After NO₃⁻ electroreduction, a certain amount of electrolyte was taken out for further quantification by ¹H NMR (Bruker AVANCE AV III 400). All analyses were performed with 128-time scans. The concentration-integral area curve was calibrated using a standard (NH₄)₂SO₄ solution. Typically, (NH₄)₂SO₄ was dissolved in 20 mL of 1 M KOH electrolyte as a series of standard (NH₄)₂SO₄ solutions with different concentrations. Subsequently, 0.1 mL of dimethyl sulfoxide-*d*₆ (DMSO-*d*₆), 0.1 mL of 6 mM 1-propanesulfonic acid 3-(trimethylsilyl) sodium salt (DSS) solution, and 0.08 mL of 6 M HCl to adjust the pH value were added into 0.32 mL of (NH₄)₂SO₄ standard solutions with different concentrations. The signal appeared at 7.23, 7.10, and 6.97 ppm were attributed to NH₄⁺. The integral areas of the signal of NH₄⁺ were used to determine the concentration of (NH₄)₂SO₄ compared with the as-known DSS reference (Supplementary Fig. 14). The yield rates of NH₃ for Co₃O₄/Cu₁-N-C at all the applied potentials determined by ¹H NMR were approximated to the results detected via the indophenol blue method (Supplementary Table 4). We have added the corresponding discussions in the revised manuscript (Page 6, lines 154-156, highlighted in yellow color).

14. The purpose of N doping in the catalyst remains unclear. Please provide an explanation.

We genuinely thank the reviewer for raising this concern. The doped N could help to anchor Cu atoms on the surface of N-doped carbon to form Cu-N bond. To further explain the role of N doping in the catalyst, we have conducted DFT calculation to monitor the adsorption energy of Cu atom on the carbon support with/without the doping of N atoms. The adsorption energy of Cu atom on N-doped carbon was -5.31 eV, which is lower than that on the carbon support (-4.51 eV), demonstrating the stronger binding of Cu atoms on N-doped carbon (Fig. R1). The uniform distribution of N throughout the whole structure was beneficial to the well-dispersed Cu single

atoms, ensuring the full exposure of Cu sites during the reaction process.

Figure R1. The structure models of Cu single atom adsorbed on (a) N-doped carbon support and (b) carbon support without the doping of N atoms. The gray, blue, and yellow spheres represent C, N, and Cu atoms, respectively.

15. Considering ammonia's solubility in alkali, was exhaust gas treatment conducted during the experimental process?

We sincerely thank this reviewer for pointing out this issue. Actually, we did set an absorption cell containing 30 mL of 1 M HCl to effectively absorb the possible escaped NH₃ from the cathode cell. After the electrolysis at each applied potential, the concentration of NH₃ in the absorption cell was lower than 1 μg mL⁻¹. In this case, the volatilization of NH₃ from the electrolytes could be negligible. We have added the corresponding discussions in the revised Methods section (Page 13, lines 364-367, highlighted in yellow color).

16. The article discusses the impact of NO₂⁻ accumulation on reaction rates. Was mechanical stirring employed during electrochemical testing? It is recommended to investigate this effect through control experiments.

We genuinely thank the reviewer for his/her valuable comment. During the electrochemical testing, the magnetic stirring of 800 rpm was employed to eliminate the effect of diffusion. As suggested, we have investigated the catalytic performance of Co₃O₄/Cu₁-N-C without the magnetic stirring in the revised manuscript. As shown in Supplementary Figure 30, at -1.0 V *vs* RHE, the FE and yield rate of NH₃ over Co₃O₄/Cu₁-N-C without the magnetic stirring were 72.4% and 87.7 mg_{NH₃} h⁻¹ cm⁻², respectively, which were much lower than that with the magnetic stirring (84.9%, 114.0 mg_{NH₃} h⁻¹ cm⁻²). This result indicates that the electroreduction of NO₃⁻ was also affected by the diffusion of reactants. We have added the corresponding discussions in the revised manuscript (Page 7, lines 248-250, highlighted in yellow color).

Reviewer #2

In this manuscript, Liu et al. report a tandem catalyst for ammonia synthesis from nitrate reduction by combining Cu single atoms and Co₃O₄ nanosheets both dispersed on nitrogen-doped carbon. The authors also employed in-situ tests and DFT calculations to clarify the synergetic role of Cu and Co₃O₄ sites, respectively, for the

simultaneously optimized binding of nitrate and nitrite ions. At present, this work seems interesting for significant alleviation of nitrite production by the designed tandem process. I will recommend this manuscript for publication after some major revisions.

We genuinely thank the reviewer for his/her valuable comments and positive recommendation.

1. After reading the overall content and research in this manuscript, I feel the presented title is not appropriate for this whole work. The authors highlighted the “simultaneously optimized adsorption” of different reaction intermediates, but I have a doubt about how to optimize the adsorption simultaneously by the catalyst $\text{Co}_3\text{O}_4/\text{Cu-N-C}$. Firstly, only two reaction-related species of $^\text{NO}_3$ and $^*\text{NO}_2$ were monitored by the combining *in-situ* spectra, as we know, the above both species are also presented in bulk electrolytes, it is not a real short-life intermediate during the whole reaction process. Besides, I cannot find the full explanation about the simultaneous optimization on the experimental spectra and theoretical calculations. In fact, a tandem mechanism for Cu-N-C and Co_3O_4 toward nitrate reduction is easier to understand for this work.*

We sincerely thank the reviewer for the constructive comments. We acknowledge that the highlight of “simultaneously optimized adsorption” might be too exaggerated despite the *in situ* spectra and theoretical calculation monitoring the conversion of $^*\text{NO}_3$ and $^*\text{NO}_2$ species. As suggested, we have revised the title to “Efficient tandem electroreduction of nitrate into ammonia through coupling Cu single atoms with adjacent Co_3O_4 ”.

2. For this synthetic process of “by adding NaBH_4 to the mixture containing Cu-N-C and $\text{Co}(\text{NO}_3)_2$ ”, I think some Co atoms have also existed on Cu-N-C support as single atoms. As shown in Figure 1c, some Co mapping signals present in Cu-N-C support rather than Co_3O_4 aggregates.

We sincerely thank the reviewer for his/her valuable comments. Indeed, the wet-chemical method to deposit Co_3O_4 on $\text{Cu}_1\text{-N-C}$ might introduce some Co atoms on $\text{Cu}_1\text{-N-C}$ supports. We have measured the loading content of Co before and after etching Co_3O_4 nanosheets in $\text{Co}_3\text{O}_4/\text{Cu}_1\text{-N-C}$. After the etching process using 1 M HCl for 2 h, the content of Co in $\text{Co}_3\text{O}_4/\text{Cu}_1\text{-N-C}$ was drastically decreased from 4.70 wt% to 0.06 wt%, confirming that majority of Co species in $\text{Co}_3\text{O}_4/\text{Cu}_1\text{-N-C}$ was in the form of Co_3O_4 . In this regard, a small amount of Co atoms might be dispersed on $\text{Cu}_1\text{-N-C}$ support. Moreover, as we discussed below in the response to comment #7, we have examined the catalytic performance of Co single atoms, suggesting that the synergetic role of Co single atoms and Cu single atoms were quite weak.

3. In Figure 1b, could the authors provide an electron diffraction pattern to better support the structural formation of Co_3O_4 nanosheets?

We sincerely thank this reviewer for his/her constructive suggestion. In the revised

manuscript, we have added the high resolution transmission electron microscopy (HRTEM) images of $\text{Co}_3\text{O}_4/\text{Cu}_1\text{-N-C}$ and the corresponding selected area electron diffraction pattern. As shown in Supplementary Figure 3, the lattice fringe distances were well indexed to (311), (400), and (440) planes of Co_3O_4 , confirming the successful formation of Co_3O_4 . We have added the corresponding discussions in the revised manuscript (Page 4, lines 83-84, highlighted in yellow color).

4. The metal contents of Copper and Cobalt in $\text{Co}_3\text{O}_4/\text{Cu-N-C}$ are suggested to be added.

We genuinely thank this reviewer for raising this issue. In the revised manuscript, we have conducted the inductively coupled plasma-optical emission spectroscopy analysis to determine the metal content of Cu and Co in $\text{Co}_3\text{O}_4/\text{Cu}_1\text{-N-C}$. The metal content of Cu and Co in $\text{Co}_3\text{O}_4/\text{Cu-N-C}$ was 0.60 wt% and 4.70 wt%, respectively. We have added the corresponding discussions in the revised manuscript (Page 4, lines 91-93, highlighted in yellow color).

5. To better clarify the structure of the catalysts, XAFS fitting at the Co K-edge for various samples should be provided.

As suggested, we have provided the EXAFS data fitting results at the Co K-edge for $\text{Co}_3\text{O}_4/\text{Cu}_1\text{-N-C}$ and $\text{Co}_3\text{O}_4/\text{N-C}$ in the revised Supplementary Information (Supplementary Fig. 9 and Table 2). The coordination number of Co-O for $\text{Co}_3\text{O}_4/\text{Cu}_1\text{-N-C}$ and $\text{Co}_3\text{O}_4/\text{N-C}$ were both about 4.0, suggesting that the coordination structure of Co_3O_4 species were similar on the two supports. We have added the corresponding discussions in the revised manuscript (Page 5, lines 118-119, highlighted in yellow color).

6. In the section of activity measurements, the authors only tested one ammonia product, all FE values for both nitrate and nitrite reduction are less than one hundred percent. Are there additional gaseous and liquid products existing? These should be tested and reported.

We sincerely thank the reviewer for his/her valuable comments. In the original version of the manuscript, we evaluated the FE for NO_2^- after NO_3^- electroreduction over $\text{Co}_3\text{O}_4/\text{Cu}_1\text{-N-C}$, $\text{Co}_3\text{O}_4/\text{Cu}_1\text{-N-C}$, and $\text{Cu}_1\text{-N-C}$, respectively (Supplementary Fig. 16). Among the three catalysts, $\text{Cu}_1\text{-N-C}$ exhibited the highest FE for NO_2^- , indicating the excessive accumulation of NO_2^- during the electroreduction of NO_3^- , which hampered the formation of NH_3 for $\text{Cu}_1\text{-N-C}$. Notably, FE for NO_2^- of $\text{Co}_3\text{O}_4/\text{Cu}_1\text{-N-C}$ was much lower than that of $\text{Cu}_1\text{-N-C}$, suggesting that the composition of Co_3O_4 with $\text{Cu}_1\text{-N-C}$ promoted the subsequent conversion of NO_2^- .

As suggested, we have quantified other liquid and gaseous products (including NH_2OH , NO , NO_2 , N_2O , N_2 , and H_2) in the revised manuscript. The amount of NH_2OH has been

determined by ^1H NMR after NH_2OH was captured by excess amount of glyoxylic acid ($\text{C}_2\text{H}_2\text{O}_3$) through oximation process (Supplementary Fig. 17). Specifically, 0.4 mL of the electrolyte after the electroreduction was mixed with 10 μL of 50% $\text{C}_2\text{H}_2\text{O}_3$ solution, followed by the addition of 0.1 mL of $\text{DMSO-}d_6$ (99%) and 0.1 mL of 6 mM DSS solution. The integral area of the signal appeared at 7.46 ppm were used to determine the concentration of NH_2OH compared with the as-known DSS reference. In this work, the amount of NH_2OH was below the detection limit for both NO_3^- and NO_2^- reduction over $\text{Co}_3\text{O}_4/\text{Cu}_1\text{-N-C}$.

Besides, nitrogen oxides including NO , NO_2 , and N_2O have been detected by an infrared gas analyzer (THA100S). The amount of NO_2 were undetectable for both NO_3^- and NO_2^- reduction. In addition, H_2 and N_2 have been detected by an on-line gas chromatograph (GC-2014) equipped with a flame ionization detector and a thermal conductivity detector. The FE for gaseous products were calculated by the following equation:

$$\text{FE} = x \times V_{\text{gas}} \times N \times F / (Q \times V_{\text{m}})$$

x : the measured mole fraction of product,

V_{gas} : the total volume of the gas (L),

N : the number of electrons transferred for the product, which is 3 for NO , 8 for N_2O , 2 for H_2 and 10 for N_2 ,

F : Faraday constant, 96485 C mol^{-1} ,

Q : total electric charge (C),

V_{m} : the molar volume of the gas, 24.5 L mol^{-1} .

We have summarized the FE for NH_2OH , NO , NO_2 , N_2O , H_2 , and N_2 for $\text{Co}_3\text{O}_4/\text{Cu}_1\text{-N-C}$ after NO_3^- and NO_2^- reduction, respectively (Supplementary Table 5). The total FE for all the liquid and gaseous products approximately equaled to 100%, demonstrating that the self-electrolysis of the catalysts rarely occurred during the electroreduction process. We have added the corresponding discussions in the revised manuscript (Page 6, lines 156-159, highlighted in yellow color).

7. Following the second comment, the possible effect of Co single atoms toward nitrate/nitrite reduction should be considered by the authors. Therefore, the ammonia activity of Co-N-C and Co single atoms on Cu-N-C catalysts also need test as a comparison.

We genuinely thank this reviewer for the valuable comments. As suggested, we have investigated the catalytic performance of Co single atoms anchored on N-doped carbon (denoted as $\text{Co}_1\text{-N-C}$) and Co single atoms anchored on $\text{Cu}_1\text{-N-C}$ (denoted as $\text{Co}_1\text{Cu}_1\text{-N-C}$). $\text{Co}_1\text{-N-C}$ and $\text{Co}_1\text{Cu}_1\text{-N-C}$ was obtained via pyrolyzing the corresponding

derivative of zeolitic imidazolate frameworks, respectively, which were prepared with the similar procedure with that of Cu₁-N-C except the metal precursor. The XRD patterns of Co₁-N-C and Co₁Cu₁-N-C both exhibited the broad peaks which were attributed to graphite carbon (Supplementary Fig. 26). The characteristic peak of metal or metal oxides were not observed, demonstrating the absence of metal or metal oxides in the two samples. Moreover, we have conducted the X-ray absorption near-edge spectroscopy (XANES) and extended X-ray absorption fine structure (EXAFS) measurements to investigate the structure of Co₁-N-C and Co₁Cu₁-N-C. The Co *K*-edge XANES spectra of Co₁-N-C and Co₁Cu₁-N-C both exhibited that the valence state of Co species were between +2 to +3 (Supplementary Fig. 27). The absence of Co-Co bond in Co₁-N-C and Co₁Cu₁-N-C further confirmed the atomic dispersion of Co species. Besides, the absence of Cu-Cu bond in the Cu *K*-edge EXAFS spectra of Co₁Cu₁-N-C confirmed the atomic dispersion of Cu species in Co₁Cu₁-N-C. These results indicate that Co₁-N-C and Co₁Cu₁-N-C catalysts have been successfully prepared.

Supplementary Figure 28 shows the FE and yield rate of NH₃ for Co₁-N-C and Co₁Cu₁-N-C toward NO₃⁻ electroreduction, respectively. As expected, Co₁Cu₁-N-C exhibited a higher catalytic activity than Co₁-N-C, which could be attributed to more active sites in Co₁Cu₁-N-C. However, the highest FE and yield rate of NH₃ for Co₁Cu₁-N-C were 68.2% and 73.9 mg_{NH3} h⁻¹ cm⁻², respectively, which were considerably lower than that for Co₃O₄/Cu₁-N-C (97.7%, 114.0 mg_{NH3} h⁻¹ cm⁻²). In this case, the contribution from the small quantity of Co single atoms in Co₃O₄/Cu₁-N-C to the enhanced performance of NO₃⁻ electroreduction could be negligible. This result suggests that synergetic role of Co single atoms and Cu single atoms were quite weak, limiting the promotion for the hydrogenation of NO₂⁻ intermediates. Besides, we have examined the catalytic performance of Co₁-N-C in NO₂⁻ electroreduction to further investigate the ability of Co single atoms to facilitate the conversion of NO₂⁻. As displayed in Supplementary Figure 29, the highest FE for NH₃ of Co₁-N-C toward NO₂⁻ electroreduction was 41.7%, which was lower than that of Co₃O₄/N-C (91.3%), indicating the sluggish conversion of NO₂⁻ over Co single atoms. We have added the corresponding discussions in the revised manuscript (Page 7, lines 175-177, highlighted in yellow color).

8. In fact, after introducing Co₃O₄ to Cu-N-C catalyst, the total number of active sites should be increased. Does it mean the high activity partly originates from the high loading of Co₃O₄ in Co₃O₄/Cu-N-C? The activity comparison of physical-mixed Co₃O₄ and Cu-N-C catalysts should be provided.

We genuinely thank the reviewer for raising these concerns. In the original manuscript, we analyzed the electrochemical surface areas of Co₃O₄/Cu₁-N-C and Cu₁-N-C by measuring the double-layer capacitance (*C*_{dl}). Co₃O₄/Cu₁-N-C possessed a larger *C*_{dl} (15.1 mF cm⁻²) than Cu₁-N-C (12.0 mF cm⁻²), demonstrating that the total number of active sites for Co₃O₄/Cu₁-N-C was increased. To assess the intrinsic activity, we normalized *j*_{NH3} based on *C*_{dl} of Co₃O₄/Cu₁-N-C and Cu₁-N-C, respectively. It is worth

noting that the normalized j_{NH_3} of $\text{Co}_3\text{O}_4/\text{Cu}_1\text{-N-C}$ were still much higher than that of $\text{Cu}_1\text{-N-C}$, indicating the higher intrinsic activity of $\text{Co}_3\text{O}_4/\text{Cu}_1\text{-N-C}$ (Fig. 2d). Besides, $\text{Co}_3\text{O}_4/\text{N-C}$ with the similar loading content of Co_3O_4 exhibited a lower activity compared with $\text{Co}_3\text{O}_4/\text{Cu}_1\text{-N-C}$. As such, the improved performance of $\text{Co}_3\text{O}_4/\text{Cu}_1\text{-N-C}$ was not attributed the high loading of Co_3O_4 .

As suggested, we have evaluated the catalytic performance of a physical mixture of $\text{Cu}_1\text{-N-C}$ and $\text{Co}_3\text{O}_4/\text{N-C}$ (denoted as mixed $\text{Cu}_1+\text{Co}_3\text{O}_4$) in 1 M KOH with 1 M NO_3^- . As shown in Supplementary Figure 23, the FE for NH_3 of mixed $\text{Cu}_1+\text{Co}_3\text{O}_4$ was 71.4% at -0.8 V vs RHE, higher than those of $\text{Cu}_1\text{-N-C}$ (54.9%) and $\text{Co}_3\text{O}_4/\text{N-C}$ (40.9%) alone, indicating the combined contributions of $\text{Cu}_1\text{-N-C}$ and $\text{Co}_3\text{O}_4/\text{N-C}$. However, the catalytic performance of mixed $\text{Cu}_1+\text{Co}_3\text{O}_4$ was inferior compared with $\text{Co}_3\text{O}_4/\text{Cu}_1\text{-N-C}$ (97.7%). In this case, the spatial couple of the adjacent sites could not be sufficiently assured by the simply physical mixing, thereby limiting the effective hydrogenation of NO_2^- into NH_3 during NO_3^- electroreduction. We have added the corresponding discussions in the revised manuscript (Page 7, lines 169-172, highlighted in yellow color).

9. After successive 20 electrolysis, the ammonia production seems decline in Figure 2h. Some examinations for spent catalyst should be measured.

We sincerely appreciate the valuable comments. In the revised manuscript, we have conducted Raman and XAFS measurements for $\text{Co}_3\text{O}_4/\text{Cu}_1\text{-N-C}$ after the electrolysis process. As shown in Supplementary Figure 18, Raman spectra of the spent $\text{Co}_3\text{O}_4/\text{Cu}_1\text{-N-C}$ shows that the peaks corresponding to E_g , F_{2g} , and A_{1g} vibration modes of Co_3O_4 remained unchanged. Besides, Cu *K*-edge and Co *K*-edge XAFS spectra for $\text{Co}_3\text{O}_4/\text{Cu}_1\text{-N-C}$ indicated that the Cu-N bonding and Co_3O_4 species were stable after the electrolysis. As such, the structure of $\text{Co}_3\text{O}_4/\text{Cu}_1\text{-N-C}$ was preserved after the electrolysis process (Supplementary Fig. 19). We have added the corresponding discussions in the revised manuscript (Page 6, lines 161-163, highlighted in yellow color).

10. The ammonia activity of N-C toward nitrate/nitrite reduction and LSV curves of $\text{Co}_3\text{O}_4/\text{Cu-N-C}$ in KOH, respectively, should also be measured as a comparison.

We sincerely thank the reviewer for raising this issue. As suggested, we have evaluated the catalytic performance of N-doped carbon toward NO_3^- and NO_2^- electroreduction in the revised manuscript, respectively. As demonstrated in Supplementary Figure 25, the highest FE and yield rate for NH_3 of N-doped carbon toward $\text{NO}_3^-/\text{NO}_2^-$ electroreduction were both below 25% and $15 \text{ mg}_{\text{NH}_3} \text{ h}^{-1} \text{ cm}^{-2}$, respectively. The catalytic properties of N-doped carbon were much lower compared with $\text{Co}_3\text{O}_4/\text{Cu}_1\text{-N-C}$ (97.7%, $114.0 \text{ mg}_{\text{NH}_3} \text{ h}^{-1} \text{ cm}^{-2}$). We have added the corresponding discussions in the revised manuscript (Page 7, lines 174-175, highlighted in yellow color).

We have collected the LSV curve of $\text{Co}_3\text{O}_4/\text{Cu}_1\text{-N-C}$ with 1 M KOH. As shown in Supplementary Figure 12, the current density of $\text{Co}_3\text{O}_4/\text{Cu}_1\text{-N-C}$ with 1 M KOH was much lower than that with 1 M KOH containing NO_3^- , implying the superior activity of $\text{Co}_3\text{O}_4/\text{Cu}_1\text{-N-C}$ toward NO_3^- electroreduction. We have added the corresponding discussions in the revised manuscript (Page 6, lines 139-141, highlighted in yellow color).

11. The detected intermediate species on Raman and FTIR spectra should be labeled in Figures.

We sincerely thank the reviewer for pointing out this. As suggested, we have labeled the detected intermediate species on Raman and FTIR spectra in the revised Figure 3.

12. The more experimental details for in-situ measurements are suggested to show in the supplementary information.

We genuinely thank this reviewer for his/her constructive suggestion. We have added more experimental details in the revised manuscript. For the *in situ* FTIR measurements, all electrochemical tests were conducted at room temperature. The background spectra of the working electrode were obtained at an open-circuit potential before the electrochemical tests. All of the spectra were collected in absorbance by averaging 32 scans at a resolution of 4 cm^{-1} . For the *in situ* Raman measurements, the electrocatalysts were deposited on fluorine tin oxide-coated glass as the working electrode. A Ag/AgCl electrode, and a Pt wire were used as the reference electrode, and counter electrode, respectively. All electrochemical tests were measured in 1 M KOH electrolyte with 1 M $\text{KNO}_3/\text{KNO}_2$ and controlled by a CHI1140C electrochemical workstation. Besides, we have provided the photographs of electrochemical cell for *in situ* measurements in the revised supplementary information (Supplementary Figs. 32 and 33). We have added the corresponding details in the revised Methods section (Page 17, lines 500-504; page 8, lines 508-516, highlighted in yellow color).

13. In Figure 3, besides the only two detected species of $^\text{NO}_3$ and $^*\text{NO}_2$ by combining Raman and FTIR spectra, why are the other intermediates absent in this work? I cannot understand these spectral results, maybe some full explanations should be needed by the authors.*

We sincerely thank the reviewer for raising these concerns. In the original manuscript, we only focused on the consumption of NO_3^- and generated NH_4^+ in the *in situ* FTIR spectra of $\text{Co}_3\text{O}_4/\text{Cu}_1\text{-N-C}$, which were detected at the wavenumbers ranging from 1360 to 1480 cm^{-1} . To further identify the intermediates during the NO_3^- electroreduction, we have carefully analyzed the original data of the *in situ* FTIR spectra. As displayed in Supplementary Figure 34, two peaks at 1541 and 1508 cm^{-1} were observed at the wavenumbers ranging from 1490 to 1550 cm^{-1} , which were assigned to the vibration band of $^*\text{NO}$ and $^*\text{NOH}$, respectively. The intensities of the two peaks increased with

the increment of applied potentials, implying the fast proceeding of NO_3^- electroreduction. We have added the corresponding details in the revised manuscript (Page 7, lines 190-192, highlighted in yellow color).

As for the Raman spectra, the low sensitivity of normal Raman impeded the detection of trace intermediates with low concentration. Besides, the broad bands of graphite carbon from 1320 to 1680 cm^{-1} could overlap with the signals of intermediates, rendering great difficult to detect the signals of intermediates for carbon-based catalysts.

14. Considering the competitive HER during the electrolysis, the adsorption strength of hydrogen atom on the surface of catalysts is also important. The authors should calculate the free energy of hydrogen adsorption on Co_3O_4 and Cu-N-C, respectively.

We sincerely thank this reviewer for his/her constructive suggestion. We have calculated the ΔG for *H adsorption over CuN_4 and Co_3O_4 slabs in the revised manuscript. As shown in Supplementary Figures 39 and 40, the ΔG for *H adsorption over CuN_4 and Co_3O_4 is 1.60 and 0.98 eV, respectively. In this regard, the adsorption of *H on CuN_4 and Co_3O_4 are both weak, which would be beneficial to the suppression of competitive HER. We have added the corresponding discussions in the revised manuscript (Page 9, lines 248-250, highlighted in yellow color).

Reviewer #3

In this manuscript, authors promote the nitrate reduction reaction (NO_3RR) performance of Cu-based catalysts by combining Cu single atoms with Co_3O_4 . NO_2^- is found to be the intermediate of NO_3RR . Cu-based electrocatalysts have poor binding affinity for NO_2^- , resulting in the sluggish absorption and reduction of NO_2^- . By dispersing Cu single atoms on Co_3O_4 , the catalyst of $\text{Co}_3\text{O}_4/\text{Cu}_1\text{-N-C}$ can be prepared and it is observed that the activity of NO_2^- reduction is highly improved. The FE and YR in electrosynthesis of ammonia from nitrate can reach 97.7% at -0.8 V vs RHE and 114.0 $\text{mg}_{\text{NH}_3}\cdot\text{h}^{-1}\cdot\text{cm}^{-2}$ at -1.0 V vs RHE. The mechanism study further provides evidence that Co_3O_4 can facilitate NO_2^- binding which increases the catalytic NO_3RR activity. This manuscript is recommended for publication after addressing the following questions.

We genuinely thank the reviewer for his/her valuable comments and positive recommendation.

Here are some suggestions to improve the manuscript:

1. Could the authors perform the control experiments on N-doped carbon without the catalyst?

We sincerely thank the reviewer for raising this issue. As suggested, we have evaluated

the catalytic performance of N-doped carbon toward NO_3^- and NO_2^- electroreduction, respectively. As demonstrated in Supplementary Figure 25, the highest FE and yield rate for NH_3 of N-doped carbon toward $\text{NO}_3^-/\text{NO}_2^-$ electroreduction were both below 25% and $15 \text{ mg}_{\text{NH}_3} \text{ h}^{-1} \text{ cm}^{-2}$, respectively. The catalytic properties of N-doped carbon were much lower compared with $\text{Co}_3\text{O}_4/\text{Cu}_1\text{-N-C}$ (97.7%, $114.0 \text{ mg}_{\text{NH}_3} \text{ h}^{-1} \text{ cm}^{-2}$). We have added the corresponding discussions in the revised manuscript (Page 7, lines 174-175, highlighted in yellow color).

2. Why do authors choose Co_3O_4 ? Could the authors examine other metal oxides to compare their ability to bind intermediates?

We sincerely thank the reviewer for the valuable comments. Considering that the multiple nitrogen-containing intermediates (e.g. *NO_3 , *NO_2 , and *NO) involved in the NO_3^- electroreduction, the moderate binding energy of intermediates serves as the key factors for efficient NO_3^- electroreduction into NH_3 . For extensively reported Cu-based catalysts, the accumulation of NO_2^- on Cu sites usually restricted the subsequent hydrogenation steps for NH_3 production. In this case, the tandem Cu-based catalysts require the combination of specific functional modules which could optimize the adsorption of NO_2^- intermediates. The previous studies have indicated that Co-based catalysts could promote the adsorption of NO_2^- from the theoretical point of view, especially CoO_x species (*Inorg. Chem.* **2023**, *62*, 15352; *Adv. Energy Mater.* **2022**, *12*, 2202105). Therefore, Co_3O_4 are considered to be a promising component for the design of an efficient tandem Cu-based catalyst towards NO_3^- electroreduction.

As suggested, we have also examined the catalytic performance of other metal oxides such as FeO_x , CuO_x , and NiO_x toward NO_2^- electroreduction to compare their ability to bind NO_2^- intermediates, respectively. The corresponding metal oxides dispersed on N-doped carbon were prepared with the similar procedure with that of $\text{Co}_3\text{O}_4/\text{N-C}$ except for the addition of $\text{Fe}(\text{NO}_3)_3 \cdot 9\text{H}_2\text{O}$, $\text{Cu}(\text{NO}_3)_2 \cdot 3\text{H}_2\text{O}$, and $\text{Ni}(\text{NO}_3)_2 \cdot 6\text{H}_2\text{O}$ as the metal precursors, respectively (denoted as $\text{FeO}_x/\text{N-C}$, $\text{CuO}_x/\text{N-C}$, and $\text{NiO}_x/\text{N-C}$, respectively). We have conducted XPS measurements for the three catalysts (Supplementary Fig. 20). The Fe $2p$ XPS spectra of $\text{FeO}_x/\text{N-C}$ exhibited two signals at 725.2 and 711.2 eV, which were attributed to $\text{Fe}^{3+} 2p_{1/2}$ and $\text{Fe}^{3+} 2p_{3/2}$, respectively. The Cu $2p$ XPS spectra of $\text{CuO}_x/\text{N-C}$ showed two peaks at 953.8 and 933.2 eV, which were assigned to $\text{Cu}^{2+} 2p_{1/2}$ and $\text{Cu}^{2+} 2p_{3/2}$, respectively. The Ni $2p$ XPS spectra of $\text{NiO}_x/\text{N-C}$ displayed two signals at 837.6 and 855.9 eV, which were assigned to $\text{Ni}^{2+} 2p_{1/2}$ and $\text{Ni}^{2+} 2p_{3/2}$, respectively. These results indicate the successful synthesis of $\text{FeO}_x/\text{N-C}$, $\text{CuO}_x/\text{N-C}$, and $\text{NiO}_x/\text{N-C}$. As shown in Supplementary Figure 21, the catalytic activity for NH_3 over the three catalysts toward the electroreduction of NO_2^- were all lower than that over $\text{Co}_3\text{O}_4/\text{N-C}$, suggesting the inferior ability of these metal oxide to facilitate the conversion of NO_2^- . We have added the corresponding discussions in the revised manuscript (Page 6, line 165; page 7, lines 166-168, highlighted in yellow color).

3. Could the authors investigate how a change in the ratio between Cu single atom and

Co₃O₄ would affect the overall NO₃RR performance?

We sincerely thank the reviewer for the constructive comments. We have investigated the influence of the ratio of Cu single atom and Co₃O₄ by controlling the amount of Co precursor in the revised manuscript. For the synthesis of Co₃O₄/Cu₁-N-C, the amount of Co(NO₃)₃·6H₂O as the precursor was 37.5 mg, denoting as 5%-Co₃O₄/Cu₁-N-C. For comparison, we set the amount of Co precursor as 22.5 and 67.5 mg, respectively. The corresponding samples were denoted as 2.5%-Co₃O₄/Cu₁-N-C and 10%-Co₃O₄/Cu₁-N-C, respectively. As shown in Supplementary Figure 22, the catalytic activity for NH₃ over the two catalysts toward the electroreduction of NO₃⁻ were both lower than that over 5%-Co₃O₄/Cu₁-N-C. The insufficient coverage of Co₃O₄ on Cu₁-N-C could not ensure the efficient conversion of the accumulated NO₂⁻, leading to undesirable performance. However, the excess amount of Co₃O₄ would completely cover the surface of Cu₁-N-C, impeding the fully exposure of Cu sites to adsorb NO₃⁻. As such, the moderate ratio between Cu single atom and Co₃O₄ are beneficial to the performance of NO₃⁻ electroreduction. We have added the corresponding discussions in the revised manuscript (Page 7, lines 168-169, highlighted in yellow color).

4. Figure 2, why is the activity of Co₃O₄/Cu₁-N-C slightly lower than that of Co₃O₄/N-C when the potential is more negative than -0.5 V vs RHE?

We sincerely thank the reviewer for raising this concern. In the electrolyte of KOH containing NO₂⁻, the current density of Co₃O₄/Cu₁-N-C were quite approach to that of Co₃O₄/N-C at lower potentials, even slightly lower at the potential ranging from -0.2 to -0.5 V vs RHE. Considering that Co₃O₄ possessed stronger binding with NO₂⁻ than Cu sites, Co₃O₄ species would serve as the main active sites at the potential ranging from -0.2 to -0.5 V vs RHE. Besides, the inferior activity of Cu sites toward NO₂⁻ electroreduction suggested that the contribution of Cu sites to the current density in Co₃O₄/Cu₁-N-C could be insignificant at the lower potentials. Given that the H₂ evolution reaction (HER) could also occur during the electrolysis, the small difference in the current density at lower potentials might be attributed to the different role of Cu₁-N-C and N-doped carbon as the supports toward the HER. Notably, as the potentials further increased, the increment of current density for Co₃O₄/Cu₁-N-C were larger than that for Co₃O₄/N-C due to the arising contribution from Cu sites toward NO₂⁻ reduction. In our work, we focused on the catalytic performance of the catalysts under the relative high current density to clearly clarify the separate functions of Cu sites and Co₃O₄ species toward the sequential conversion of NO₃⁻ to NO₂⁻ and NO₂⁻ to NH₃, respectively.

5. Page 4, Line 93. The lattice fringes with an interplanar spacing of 0.201 nm were ascribed to the (400) facet of Co₃O₄. This number is different from reported Co₃O₄. Could the authors explain this difference? Does this difference have an effect on NO₂⁻ adsorption?

We genuinely thank this reviewer for the valuable comments. To further clarify the

structure of Co_3O_4 , we have provided the selected area electron diffraction pattern of Co_3O_4 nanosheets in the revised manuscript. The lattice fringe distances were well indexed to (311), (400), and (440) planes of Co_3O_4 , confirming the successful formation of Co_3O_4 (Supplementary Fig. 3). In our work, the thickness and size of Co_3O_4 nanosheets were quite small, delivering great difficulty to calibrate all the crystal facets such as (311) and (440). The (400) facet of Co_3O_4 with a lattice spacing of 0.20 nm was also observed in previously reported works (*Small*, **2018**, *14*, 1800225; *Adv. Funct. Mater.*, **2023**, *33*, 2210945). Moreover, we have employed Co_3O_4 (311) and Co_3O_4 (440) slabs as the models to evaluate the adsorption of NO_2^- on different facets. The Gibbs free-energy changes (ΔG) for NO_2^- adsorption over Co_3O_4 (311) and Co_3O_4 (440) slabs were 0.97 and 0.78 eV, respectively, which were lower than that over CuN_4 (1.31 eV), indicating the stronger binding of NO_2^- over Co_3O_4 regardless of the facet (Fig. R2).

Figure. R2. The structure models of NO_2^- adsorbed on (a) Co_3O_4 (311) and (b) Co_3O_4 (440) slabs. The blue, red, and purple spheres represent N, O, and Co atoms, respectively.

6. Page 6, Line 143. $\text{Co}_3\text{O}_4/\text{Cu}_1\text{-N-C}$ achieved the maximum FE for NH_3 of 97.7% at -0.8 V vs RHE. Could the authors quantify the other products under this condition, which could make up for the remaining FE? (e.g. NO_2^- , NH_2OH , NO , NO_2 , N_2O , N_2 , and/or H_2).

We sincerely thank the reviewer for his/her valuable comments. In the original version of the manuscript, we evaluated the FE for NO_2^- after NO_3^- electroreduction over $\text{Co}_3\text{O}_4/\text{Cu}_1\text{-N-C}$, $\text{Co}_3\text{O}_4/\text{Cu}_1\text{-N-C}$, and $\text{Cu}_1\text{-N-C}$ (Supplementary Fig. 16). Among the three catalysts, $\text{Cu}_1\text{-N-C}$ exhibited the highest FE for NO_2^- , indicating the excessive accumulation of NO_2^- during the electroreduction of NO_3^- , which hampered the formation of NH_3 for $\text{Cu}_1\text{-N-C}$. Notably, FE for NO_2^- of $\text{Co}_3\text{O}_4/\text{Cu}_1\text{-N-C}$ was much lower than that of $\text{Cu}_1\text{-N-C}$, suggesting that the composition of Co_3O_4 with $\text{Cu}_1\text{-N-C}$ promoted the subsequent conversion of NO_2^- .

As suggested, we have quantified other liquid and gaseous products (including NH_2OH , NO , NO_2 , N_2O , N_2 , and H_2) in the revised manuscript. The amount of NH_2OH has been determined by ^1H NMR after NH_2OH was captured by excess amount of glyoxylic acid ($\text{C}_2\text{H}_2\text{O}_3$) through oximation process (Supplementary Fig. 17). Specifically, 0.4 mL of the electrolyte after the NO_3^- electroreduction was mixed with 10 μL of 50% $\text{C}_2\text{H}_2\text{O}_3$

solution, followed by the addition of 0.1 mL of DMSO-*d*₆ (99%) and 0.1 mL of 6 mM DSS solution. The integral area of the signal appeared at 7.46 ppm were used to determine the concentration of NH₂OH compared with the as-known DSS reference. In this work, the amount of NH₂OH was below the detection limit for NO₃⁻ reduction over Co₃O₄/Cu₁-N-C.

Besides, nitrogen oxides including NO, NO₂, and N₂O have been detected by an infrared gas analyzer (THA100S). The amount of NO₂ were undetectable for both NO₃⁻ and NO₂⁻ reduction. In addition, H₂ and N₂ have been detected by an on-line gas chromatograph (GC-2014) equipped with a flame ionization detector and a thermal conductivity detector. The FE for gaseous products were calculated by the following equation:

$$FE = x \times V_{\text{gas}} \times N \times F / (Q \times V_{\text{m}})$$

x: the measured mole fraction of product,

*V*_{gas}: the total volume of the gas (L),

N: the number of electrons transferred for the product, which is 3 for NO, 8 for N₂O, 2 for H₂ and 10 for N₂,

F: Faraday constant, 96485 C mol⁻¹,

Q: total electric charge (C),

*V*_m: the molar volume of the gas, 24.5 L mol⁻¹.

We have summarized the FE for NH₂OH, NO, NO₂, N₂O, H₂, and N₂ for Co₃O₄/Cu₁-N-C after NO₃⁻/NO₂⁻ reduction, respectively (Supplementary Table 5). The total FE for all the liquid and gaseous products approximately equaled to 100%, demonstrating that the self-electrolysis of the catalysts rarely occurred during the electroreduction process. We have added the corresponding discussions in the revised manuscript (Page 6, lines 156-159, highlighted in yellow color).

7. The best yield rate is obtained at -1.0 V vs RHE. Could the authors comment on the competitive reaction (e.g. HER) under this (rather negative) applied potential?

We genuinely thank this reviewer for the valuable comments. Despite that the best yield rate of NH₃ was obtained at -1.0 V vs RHE, the FE for NH₃ was 84.9%, which was slightly lower than that at -0.8 V vs RHE (97.7%). To further evaluate the competitive HER at -1.0 V vs RHE, we have measured the FE for H₂ over Co₃O₄/Cu₁-N-C. As we expected, the FE for H₂ was 11.7%, which was higher than that at -0.8 V vs RHE (0.6%). In this regard, the competitive HER was gradually increasing as the applied potential increased.

8. Page 7, Line 190. The authors calculated the rate constants for NO_3^- and NO_2^- electroreduction (k_1 & k_2). In Figure S8, the author excluded the possible ammonia contamination from the self-electrolysis of the catalyst under NO_3^- conditions. Likewise, could the authors perform an experiment to exclude self-catalysis under NO_2^- conditions?

We genuinely thank this reviewer for pointing out this issue. In the original manuscript, the electrolysis of $\text{Co}_3\text{O}_4/\text{Cu}_1\text{-N-C}$ was conducted in 1 M KOH electrolyte to exclude possible ammonia contamination from the self-electrolysis (Supplementary Fig. 24). Specifically, after the electrochemical test proceeded in 1 M KOH electrolyte at -0.8 V vs RHE, the yield of NH_3 was below detection limit. As such, the NH_3 production under $\text{NO}_3^-/\text{NO}_2^-$ conditions both did not derive from the self-electrolysis of the catalyst. Besides, the total FE for all the liquid and gaseous products after NO_2^- electroreduction approximately equaled to 100%, demonstrating that the self-electrolysis of the catalysts rarely occurred during the electroreduction process (Supplementary Table 5).

9. Page 16, Figure 3f and 3d, what are the two peaks around 1380 cm^{-1} and 1620 cm^{-1} ? Why do they look different for the two catalysts? Why do the signals change when the focal plane of the Raman laser is moved further away from the surface of the catalysts? Also, why did the authors put emphasis on this laser-distance experiment? Is there any specific reason?

We sincerely thank this reviewer for raising these issues. The two peaks around 1380 and 1620 cm^{-1} in Figures 3d and 3f were assigned to the D band and G band of graphite carbon, respectively. Figure 3d shows the *in situ* Raman spectra for $\text{Cu}_1\text{-N-C}$ at different potentials ranging from OCP to -1.0 V vs RHE in 1 M NO_3^- . Figure 3f shows the *in situ* Raman spectra for $\text{Co}_3\text{O}_4/\text{Cu}_1\text{-N-C}$ at -0.8 V vs RHE when the laser beam was positioned 0 to 200 μm away from the surface of catalysts. The signals of graphite carbon in the two figures looked different owing to the distinct scale of the vertical axis. To clearly show the variation of peaks for NO_2^- near the surface of the catalysts, we adjusted the scale of the vertical axis in Figure 3f to be much smaller than that in Figure 3d, giving rise to the difference in appearance of signals from graphite carbon.

As the focal plane of the Raman laser is moved further away from the surface of the catalysts, the laser beam is focused into liquid instead of on the catalysts. Due to the increment of the distance between the focal plane of laser and the surface of catalysts, the signal intensity of graphite carbon for the catalysts gradually decreased. This laser-distance experiment was conducted to further probe the variation of local concentration for NO_2^- near the surface of the catalysts. Based on the integrated areas of NO_2^- and NO_3^- , the local concentration of NO_2^- was estimated near the surface of $\text{Cu}_1\text{-N-C}$ and $\text{Co}_3\text{O}_4/\text{Cu}_1\text{-N-C}$, respectively. The much lower concentration of NO_2^- for $\text{Co}_3\text{O}_4/\text{Cu}_1\text{-N-C}$ than $\text{Cu}_1\text{-N-C}$ further manifested the facilitated reduction of NO_2^- with the favor of Co_3O_4 . We have added the corresponding discussions in the revised manuscript (Page 8, lines 207-208, highlighted in yellow color).

10. Page 17, Figure 4c. In order to compare the Gibbs Free Energy of CuN_4 and Co_3O_4 with the intermediate, the free energy diagram of NO_3^- and NO_2^- intermediate is separated into two parts with a blue and green background, and both begin at 0 eV. While the NO_3RR is a continuous process, is it reasonable to merge these two graphs into one? Could the authors explain more in-depth as to why the figure is drawn this way? Is it to show that CuN_4 (blue line) is the better catalyst for the NO_3^- -to- NO_2^- process (blue region) while Co_3O_4 (green line) is the better catalyst for the NO_2^- -to- NH_3 process (green region)?

We sincerely thank this reviewer for the constructive comments. Based on the experimental results, the electroreduction of NO_3^- to NH_3 was proposed to be an independent two-step process including an initial reduction of NO_3^- to NO_2^- and the followed reduction of NO_2^- to NH_3 . To elucidate the separate functions of Cu sites and Co_3O_4 species for the conversion of NO_3^- and NO_2^- , respectively, we employed CuN_4 and Co_3O_4 slab as the models to conduct the DFT calculation. The free energy diagram was separated into two parts to clearly reflect the tandem catalysis proceeded on the separate sites. As displayed in Figure 4c, CuN_4 is the better catalyst for the NO_3^- -to- NO_2^- process (blue region) while Co_3O_4 (green line) is the better catalyst for the NO_2^- -to- NH_3 process (green region).

REVIEWER COMMENTS

Reviewer #1 (Remarks to the Author):

Thanks to the authors for their efforts in addressing the majority of the reviewer's concerns. However, it is worth noting that this work lacks a clear assertion of its innovativeness. Copper-cobalt-based tandem catalysts have been previously reported, and many of these catalysts have effectively overcome the issue of excessive NO₂⁻ accumulation [1-6]. In addition, the performance demonstrated by the authors in this study falls short compared to other tandem catalysts. The achieved ampere-level current in this work requires a substantial overpotential (> -0.6 V versus RHE), whereas literature reports indicate that similar currents can be obtained at much lower overpotentials (-0.2~-0.4 V versus RHE)[7-9]. Therefore, the author's claim regarding improved reaction kinetics through this tandem catalyst design by combining Cu single atoms catalysts with adjacent Co₃O₄ nanosheets remains insufficiently substantiated.

Therefore, the contribution of this work to the development of the field appears to be rather limited and this work lacks sufficient innovation to warrant recommendation for publication in this high-impact journal. It may be more suitable for submission to other journals.

Reference:

- [1] He W., et al. Splicing the active phases of copper/cobalt-based catalysts achieves high-rate tandem electroreduction of nitrate to ammonia. *Nat. Commun.*, 2022, 13,1129.
- [2] Niu Z., et al. Bifunctional copper-cobalt spinel electrocatalysts for efficient tandem-like nitrate reduction to ammonia. *Chem. Eng. J.*, 2022, 450, 138343.
- [3] Li P., et al. Pulsed nitrate-to-ammonia electroreduction facilitated by tandem catalysis of nitrite intermediates. *J. Am. Chem.Soc.*, 2023, 145, 6471.
- [4] Song M., et al. Fe and Cu Double-Doped Co₃O₄ Nanorod with Abundant Oxygen Vacancies: A High-Rate Electrocatalyst for Tandem Electroreduction of Nitrate to Ammonia. *Inorg. Chem.*, 2023, 62, 16641.
- [5] Wu R., et al. Selective tandem electroreduction of nitrate to nitrogen via copper–cobalt based bimetallic hollow nanobox catalysts. *Environ. Sci. Nano*, 2023, 10, 2332.
- [6] Wang W., et al. Synergy between Cu and Co in a Layered Double Hydroxide Enables Close to 100% Nitrate-to-Ammonia Selectivity. *J. Am. Chem. Soc.*, 2023, 145, 26678.
- [7] Fang, J., et al. Ampere-level current density ammonia electrochemical synthesis using CuCo nanosheets simulating nitrite reductase bifunctional nature. *Nat. Commun.*, 2022, 13, 7899.
- [8] Yu, W., et al. Laser-controlled tandem catalytic sites of CuNi alloy with ampere-level electrocatalytic nitrate to ammonia activity for Zn–nitrate battery. *Energy Environ. Sci.*, 2023, 16, 2991.

[9] Zhang X., et al. Tandem Nitrate Electroreduction to Ammonia with Industrial-Level Current Density on Hierarchical Cu Nanowires Shelled with NiCo-Layered Double Hydroxide. ACS Catal., 2023, 13, 14670.

Reviewer #2 (Remarks to the Author):

The authors have addressed most of my concerns. I recommend to accept it without further revision.

Reviewer #3 (Remarks to the Author):

The authors have addressed all my concerns and questions. I have no further comments, and I recommend this manuscript be accepted.

Just a suggestion, the authors could consider including more literature references to support their response to reviewers' question 1. In particular, recently there are quite a handful of papers published on Nat Commun, JACS, and Angew detailing how bimetallic active sites enhances NO₃RR synergistically. Also, there are quite a few SERS studies back in the days on JACS and Nano Energy describing the binding interactions between *NO₃ and electrode surfaces. I don't want to mention names/research group or particular papers here, but I trust that the authors will be able to find suitable and relevant references to further support their claims and hypotheses.

Point-by-point response to reviewer comments

Manuscript ID: NCOMMS-23-47388A

MS Type: Research Article

Title: "Efficient tandem electroreduction of nitrate into ammonia through coupling Cu single atoms with adjacent Co₃O₄"

Reviewer #1

Thanks to the authors for their efforts in addressing the majority of the reviewer's concerns. However, it is worth noting that this work lacks a clear assertion of its innovativeness. Copper-cobalt-based tandem catalysts have been previously reported, and many of these catalysts have effectively overcome the issue of excessive NO₂⁻ accumulation [1-6]. In addition, the performance demonstrated by the authors in this study falls short compared to other tandem catalysts. The achieved ampere-level current in this work requires a substantial overpotential (> -0.6 V versus RHE), whereas literature reports indicate that similar currents can be obtained at much lower overpotentials (-0.2~-0.4 V versus RHE)[7-9]. Therefore, the author's claim regarding improved reaction kinetics through this tandem catalyst design by combining Cu single atoms catalysts with adjacent Co₃O₄ nanosheets remains insufficiently substantiated. Therefore, the contribution of this work to the development of the field appears to be rather limited and this work lacks sufficient innovation to warrant recommendation for publication in this high-impact journal. It may be more suitable for submission to other journals.

Reference:

- [1] He W., et al. Splicing the active phases of copper/cobalt-based catalysts achieves high-rate tandem electroreduction of nitrate to ammonia. Nat. Commun., 2022, 13, 1129.*
- [2] Niu Z., et al. Bifunctional copper-cobalt spinel electrocatalysts for efficient tandem-like nitrate reduction to ammonia. Chem. Eng. J., 2022, 450, 138343.*
- [3] Li P., et al. Pulsed nitrate-to-ammonia electroreduction facilitated by tandem catalysis of nitrite intermediates. J. Am. Chem. Soc., 2023, 145, 6471.*
- [4] Song M., et al. Fe and Cu Double-Doped Co₃O₄ Nanorod with Abundant Oxygen Vacancies: A High-Rate Electrocatalyst for Tandem Electroreduction of Nitrate to Ammonia. Inorg. Chem., 2023, 62, 16641.*
- [5] Wu R., et al. Selective tandem electroreduction of nitrate to nitrogen via copper-cobalt based bimetallic hollow nanobox catalysts. Environ. Sci. Nano, 2023, 10, 2332.*
- [6] Wang W., et al. Synergy between Cu and Co in a Layered Double Hydroxide Enables Close to 100% Nitrate-to-Ammonia Selectivity. J. Am. Chem. Soc., 2023, 145, 26678.*
- [7] Fang, J., et al. Ampere-level current density ammonia electrochemical synthesis using CuCo nanosheets simulating nitrite reductase bifunctional nature. Nat. Commun., 2022, 13, 7899.*
- [8] Yu, W., et al. Laser-controlled tandem catalytic sites of CuNi alloy with ampere-level electrocatalytic nitrate to ammonia activity for Zn-nitrate battery. Energy Environ. Sci., 2023, 16, 2991.*
- [9] Zhang X., et al. Tandem Nitrate Electroreduction to Ammonia with Industrial-Level*

Current Density on Hierarchical Cu Nanowires Shelled with NiCo-Layered Double Hydroxide. ACS Catal., 2023, 13, 14670.

We sincerely appreciate the insightful comments from the reviewer. Although copper-cobalt-based tandem catalysts have been previously reported, the comprehension of the real role of Cu sites and Co sites in the reaction process are really diverse, and even opposite. For example, Cu-based sites were usually considered as the sites to bind NO_3^- (*Nat. Commun.*, **2022**, *13*, 1129; *Inorg. Chem.*, **2023**, *62*, 16641; *J. Am. Chem. Soc.*, **2023**, *145*, 26678). Nevertheless, the carbon fiber supports in $\text{CuCo}_2\text{O}_4/\text{CFs}$ were reported to promote the conversion of NO_3^- to NO_2^- whereas the Cu and Co sites in CuCo_2O_4 nanoparticles served as the active sites for $^*\text{NO}_2$ (*Chem. Eng. J.*, **2022**, *450*, 138343). With regard to the real role of Co-based sites, some works regarded the oxygen vacancy in Co_3O_4 as the active sites for $^*\text{NO}_2$ (*Inorg. Chem.*, **2023**, *62*, 16641) while some research reported that Co sites facilitated $^*\text{H}$ production to selectively reduce NO_3^- to N_2 (*Environ. Sci. Nano*, **2023**, *10*, 2332). The reported tandem catalysts usually are typical complex assemblies, where sites are difficult to separate and would be interfered by the electron transfer, posing significant challenges to identifying the real role of different active sites.

Given that single-atom catalysts with well-defined structures could serve as an ideal platform for investigating the mechanism, it is highly promising to construct a tandem catalyst by introducing spatially separated active sites into single-atom catalysts. In our work, we constructed spatially separated and electronically decoupled dual sites by depositing Co_3O_4 on $\text{Cu}_1\text{-N-C}$, which is more applicable for investigating separate functions of Cu single atoms and Co_3O_4 . The spatially separated Cu sites and Co_3O_4 species were uniformly distributed throughout the whole structure, constituting the adjacent catalytic centers. Combining with *in situ* Raman experiments, adsorption experiment, and theoretical calculation, we clarified the separate functions of Cu single atoms and Co_3O_4 toward the conversion of NO_3^- to NO_2^- and NO_2^- to NH_3 , respectively. It is evidenced that the couple of adjacent Cu sites and Co_3O_4 species benefited the transfer of NO_2^- intermediates, promoting the hydrogenation step to NH_3 production.

Besides, the reviewer raised a concern that $\text{Co}_3\text{O}_4/\text{Cu}_1\text{-N-C}$ required a more negative potential (>-0.6 V *vs* RHE) to achieve ampere-level current than previously reported catalysts (-0.2 V \sim -0.4 V *vs* RHE). It should be pointed out that the previously reported works employed Ni foam, Cu foil, and Cu foam as the working electrode, respectively. The metal-based catalysts were prepared by electrodeposition method or laser irradiation technology on the metal substrates, which were endowed with large electrochemically active surface area. For instance, the values of double-layer capacitance (C_{dl}) for CuNi NPs/NF were 35 mF cm^{-2} (*Energy Environ. Sci.*, **2023**, *16*, 2991). In our work, carbon paper coated with $\text{Co}_3\text{O}_4/\text{Cu}_1\text{-N-C}$ was used as the working electrode with a C_{dl} of 15.1 mF cm^{-2} , which was smaller than the reported CuNi NPs/NF. The current density normalized by C_{dl} of $\text{Co}_3\text{O}_4/\text{Cu}_1\text{-N-C}$ was 45.7 mA mF^{-1} at -0.5 V *vs* RHE, which was higher than that of CuNi NPs/NF (35.3 mA mF^{-1} at -0.48 V *vs* RHE).

In this case, Co₃O₄/Cu₁-N-C showed a superior intrinsic activity toward NO₃⁻ electroreduction. Indeed, these reported catalysts realized electroreduction of NO₃⁻ at a lower potential. However, the intrinsic reason for the promoted performance of reported catalysts was attributed to the sufficient supply of *H provided by the metallic Co sites or Ni sites (*Nat. Commun.*, **2022**, *13*, 7899; *Energy Environ. Sci.*, **2023**, *16*, 2991), which was entirely different from our work. We aimed at constructing spatially separated and electronically decoupled dual sites to clarify the real role of dual sites in NO₃⁻ electroreduction. The separate functions of Cu single atoms and Co₃O₄ toward the conversion of NO₃⁻ to NO₂⁻ and NO₂⁻ to NH₃, respectively, were evidenced by *in situ* Raman experiments, adsorption experiment, and theoretical calculation. The couple of adjacent Cu sites and Co₃O₄ species benefited the transfer of NO₂⁻ intermediates, promoting the hydrogenation step to NH₃ production. This work not only develops an attractive tandem catalyst by constructing spatially separated and electronically decoupled sites but also offers a legible profile for deepening the understanding of tandem catalysis.

Reviewer #2

The authors have addressed most of my concerns. I recommend to accept it without further revision.

We sincerely thank the reviewer for his/her positive comments.

Reviewer #3

The authors have addressed all my concerns and questions. I have no further comments, and I recommend this manuscript be accepted.

We sincerely thank the reviewer for his/her positive comments.

*Just a suggestion, the authors could consider including more literature references to support their response to reviewers' question 1. In particular, recently there are quite a handful of papers published on *Nat Commun*, *JACS*, and *Angew* detailing how bimetallic active sites enhances NO₃RR synergistically. Also, there are quite a few SERS studies back in the days on *JACS* and *Nano Energy* describing the binding interactions between *NO₃ and electrode surfaces. I don't want to mention names/research group or particular papers here, but I trust that the authors will be able to find suitable and relevant references to further support their claims and hypotheses.*

We appreciate the reviewer's valuable comments. As suggested, we have added more relevant references to support our concepts in the revised manuscript.